

# Underwater light climate and wavelength dependence of microalgae photosynthetic parameters in a temperate sea

Monica Michel-Rodriguez[1], Sebastien Lefebvre[1], Muriel Crouvoisier[1], Xavier Mériaux[2] and Fabrice Lizon[1]

[1] Univ. Lille, CNRS, Univ. Littoral Côte d'Opale, UMR 8187 LOG, Laboratoire d'Océanologie et de Géosciences, Lille, France
[2] Univ. Littoral Côte d'Opale, CNRS, Univ. Lille, UMR 8187—LOG—Laboratoire d'Océanologie et de Géosciences, Wimereux, France

## ABSTRACT

Studying how natural phytoplankton adjust their photosynthetic properties to the quantity and quality of underwater light (*i.e.* light climate) is essential to understand primary production. A wavelength-dependent photoacclimation strategy was assessed using a multi-color pulse-amplitude-modulation chlorophyll fluorometer for phytoplankton samples collected in the spring at 19 locations across the English Channel. The functional absorption cross section of photosystem II, photosynthetic electron transport ($PET_\lambda$) parameters and non-photochemical quenching were analyzed using an original approach with a sequence of three statistical analyses. Linear mixed-effects models using wavelength as a longitudinal variable were first applied to distinguish the fixed effect of the population from the random effect of individuals. Population and individual trends of wavelength-dependent $PET_\lambda$ parameters were consistent with photosynthesis and photoacclimation theories. The natural phytoplankton communities studied were in a photoprotective state for blue wavelengths (440 and 480 nm), but not for other wavelengths (green (540 nm), amber (590 nm) and light red (625 nm)). Population-detrended $PET_\lambda$ values were then used in multivariate analyses (partial triadic analysis and redundancy analysis) to study ecological implications of $PET_\lambda$ dynamics among water masses.

Two wavelength ratios based on the microalgae saturation parameter $E_k$ (in relative and absolute units), related to the hydrodynamic regime and underwater light climate, clearly confirmed the physiological state of microalgae. They also illustrate more accurately that natural phytoplankton communities can implement photoacclimation processes that are influenced by *in situ* light quality during the daylight cycle in temporarily and weakly stratified water. Ecological implications and consequences of $PET_\lambda$ are discussed in the context of turbulent coastal ecosystems.

Corresponding authors
Monica Michel-Rodriguez, monica.michel-rodriguez@univ-lille.fr
Fabrice Lizon, fabrice.lizon@univ-lille.fr

## INTRODUCTION

In nature, phytoplankton must respond to multiple variations in the quantity and quality of light (*i.e.* light climate) at different temporal (from day to year) and spatial (from environmental coastal gradients to large hydrological structures) scales (*MacIntyre, Kana & Geider, 2000*; *Dubinsky & Schofield, 2010*). It is well known that microalgae have a strong ability to photoregulate, photoacclimate and photoadapt to these variations, as demonstrated by many articles and reviews (*e.g. Anning et al., 2000*; *Dubinsky & Stambler, 2009*). *Kirk (2011)* reviewed these photobiological processes and defined them as ecological strategies, highlighting the role of the light climate. Microalgae adapt to variability in the light climate through phylogenetic adaptations and ontogenetic acclimation. Evidence of phylogenetic adaptation has existed since *Engelmann (1883)* developed chromatic adaptation theory and has experienced some controversy (*e.g. Bidigare et al., 1990*; *Falkowski & LaRoche, 1991*). Pigment composition and thus cell absorption spectra, which determine light-use efficiency, have evolved to match the spectral characteristics of the prevailing light in a water mass. Ontogenetic acclimation in response to light conditions at the time of cell growth and development may modify a species' pigment composition and photosynthetic functioning, thus significantly influencing wavelength-dependent light absorption. Physiological state and photosynthetic properties of phytoplankton can be studied by using photosynthetic light-response (PE) curves to estimate photosynthetic activity as light levels increase (*Platt & Jassby, 1976*). Light-use efficiency (initial slope, $\alpha$, see Table 1 for symbols, abbreviations and definitions related to photosynthetic parameters and variable fluorescence measures) and maximum photosynthetic rate ($\rho_{max}$) parameters of PE curves are the two main parameters traditionally used to investigate biophysical, biochemical and metabolic processes that influence photosynthesis (*MacIntyre et al., 2002*; *Falkowski & Raven, 2007*) in response to variations in the light climate. Understanding better the response of cells to potential light stress in surface water also requires studying the distribution of light energy between the photochemical and non-photochemical pathways, which includes thermal dissipation of excess absorbed light energy (*Lavaud, 2007*). These processes are well documented for diatoms (*Brunet & Lavaud, 2010*) and can be studied easily by quantifying the light response (E from light Energy) of non-photochemical quenching (NPQ, *Serôdio & Lavaud, 2011*).

Measuring the light absorption capacity of microalgae is essential to estimate the survival and production capacity of cells, and for ecologists to assess photosynthetic activity and primary production. To this end, a new generation of commercial fluorometers (*e.g.* multi-color pulse-amplitude-modulation (PAM) chlorophyll fluorometer (Heinz Walz GmbH, Germany), mini-FIRe (*Gorbunov et al., 2020*)) has been designed to study wavelength dependence of photosynthetic electron transport ($PET_\lambda$) in relation to the light absorption capacity and/or to focus on general photosynthetic activity of phytoplankton groups in a given ecosystem (*e.g.* a fast repetition rate fluorometer or FRRf (Chelsea Technologies Group Ltd., United Kingdom), FFL-40 (Photon Systems Instruments, Czech Republic). In limnology and oceanography, nearly all studies that included *in vivo* chlorophyll *a* (Chla) variable fluorescence have measured
**Table 1 Abbreviations and definitions.**

| Abbreviation | Definition | Unit |
|---|---|---|
| r.α | Maximum light-use efficiency | Electrons quanta$^{-1}$ |
| α(II) | α related to absolute absorption of PSII | Electrons quanta$^{-1}$ |
| CCA | Complementary Chromatic Adaptation (*Kehoe & Gutu, 2006*) | unitless |
| CDOM | Colored Disolved Organic Matter | not reported here |
| DIN | Dissolved Inorganic Nitrogen | µmol L$^{-1}$ |
| EC | English Channel | unitless |
| $E_k$ | Light saturation coefficient | quanta m$^{-2}$ s$^{-1}$ |
| $E_k$(II) | $E_k$ related to absolute absorption of PSII | quanta (PS II s)$^{-1}$ |
| $E_{op}$ | PAR at ETR$_{max}$ | µmol quanta m$^{-2}$ s$^{-1}$ |
| $E_{op}$(II) | PAR at ETR$_{max}$(II) | quanta (PS II s)$^{-1}$ |
| $E_{avg}$ | Vertically averaged light intensity | µmol quanta m$^{-2}$ s$^{-1}$ |
| ETR(II) | ETR related to absolute absorption of PSII | electrons (PS II s)$^{-1}$ |
| ETR$_{max}$(II) | ETR$_{max}$ related to absoluted absorption of PSII | electrons (PS II s)$^{-1}$ |
| FCP | Fucoxanthin-chlorophyll a/c-binding antenna pigment-protein complex of diatoms | unitless |
| $F_v/F_m$ | Maximum quantum yield of PSII determined after 2.5 h of dark acclimation | unitless |
| $K_{d(PAR)}$ | PAR extinction coefficient also known as diffuse attenuation coefficient | m$^{-1}$ |
| O-I$_1$ | Photochemical phase of fast fluorescence rise (*Schreiber, 2004*) | unitless |
| NPQ | Non-photochemical fluorescence quenching | unitless |
| NPQ$_{1200}$ | NPQ calculated from fitted NPQ *vs.* PAR curves at PAR = 1,200 µmol quanta m$^{-2}$ s$^{-1}$ | unitless |
| NPQ$_{300}$ | NPQ calculated from fitted NPQ *vs.* PAR curves at PAR = 300 µmol quanta m$^{-2}$ s$^{-1}$ | unitless |
| PAR | Photosynthetically active radiation | µmol quanta m$^{-2}$ s$^{-1}$ |
| PE curve | Production *vs.* Irradiance (Energy) curve | unitless |
| PET | Photosynthetic electron transport | unitless |
| ρmax | General acronym for light-saturated maximum rate from PE curve | Not reported here |
| PSII | Photosystem II | unitless |
| r.ETR | Relative electron transport rate | µmol electrons m$^{-2}$ s$^{-1}$ |
| r.ETR$_{max}$ | Maximum electron transport rate | µmol electrons m$^{-2}$ s$^{-1}$ |
| Sigma(II)$_\lambda$ | Wavelength-dependent cross section of PSII | nm$^2$ |
| TSS | Time since sunrise | h |
| Y(II) | Effective quantum yield of PSII | unitless |
| $Z_{eu}$ | Depth of the euphotic layer | m |
| Zumixl | Depth of the upper mixed layer | m |

Note:
Abbreviations and definitions of variable fluorescence measurements and photosynthetic parameters (in relative and absolute units) used in the study.

direct light absorption capacity and photosynthesis for only one color of light. In most PAM techniques blue and red wavelengths (±470 and 650 nm) were typically used for measuring lights. In most recent FRRf studies, blue wavelength (±450 nm) was generally used because it is one of the main spectral bands absorbed by Chla (400–500 nm), the most common and abundant photosynthetic pigment, and the dominant color in the marine environment (*Schreiber, Klughammer & Kolbowski, 2012*). One exception is the recent study of *Houliez et al. (2017)*, who performed the first *in situ* measurements of light

absorption capacity and photosynthetic yield with blue (458 nm) and amber (593 nm) lights in the Baltic Sea. However, since it focused on the specific problem of measuring fluorescence rise in cyanobacteria, its results cannot be generalized to other phytoplankton groups.

Many studies have shown that pigment absorbance by microalgae is strongly correlated with the spectral transmittance of water and its components (*Hickman et al., 2010*; *Lawrenz & Richardson, 2017*). Colored dissolved organic matter (CDOM) and suspended particle concentrations can dramatically change the quantity and, especially, quality of light in coastal water (*Kirk, 2011*). To help Chla absorb light energy at different wavelengths, microalgae have a variety of accessory pigments. For example, the ecological success of diatoms is due to their pigment signature (*Falkowski & Knoll, 2007*), which includes Chla, Chlc and fucoxanthin (which expand the spectral absorption band to 580 nm), along with β-carotene and the xanthophylls involved in photoprotection (*Brunet & Lavaud, 2010*; *Jeffrey, Wright & Zapata, 2011*). This diversity of pigments enables brown algae, such as diatoms, to be more effective than green or red algae (*Lavaud, 2007*) in turbulent systems or in the mixed layer of the coastal ocean. However, the photosynthetic apparatus can acclimate to variations in light climate by changing cell pigment concentrations and/or ratios (*MacIntyre et al., 2002*). This can change the shape of the light-absorption spectrum and influence the efficiency of photosynthesis (*Barlow et al., 2013*, *2017*). When light decreases, pigment concentrations usually increase in cells during growth, with or without wavelength-dependent changes in light absorption (*Falkowski & LaRoche, 1991*). In addition, pigment concentration can also increase due to an increase in the size and/or number of photosynthetic units (*i.e.* antennas containing light-harvesting pigments) (*Dubinsky & Stambler, 2009*) depending on the phytoplankton group and ecosystem. Under high light conditions, cells increase the reaction center number with a smaller antenna size, inducing higher values of $\rho_{max}$. On the opposite, under low light conditions, cells increase their antenna size, inducing higher values of $\alpha$. However, light harvesting by cells is not always correlated with pigment concentration due to mutual shading of the increasing density of pigment molecules (*i.e.* the "package effect" (*Bidigare et al., 1990*)).

In response to changes in light color, an effective photoacclimation mechanism was observed in cyanobacteria that involves regulating "complementary chromatic adaptation" (CCA) (*Kehoe & Gutu, 2006*). CCA involves strong restructuring of photosynthetic antennas through pigment concentrations, including pigment-binding antenna proteins. Diatoms have fewer flexible binding proteins, such as fucoxanthin-chlorophyll a/c-binding antenna pigment-proteins complexe (FCPs), than cyanobacteria with which to perform classic CCA; however, diatom fucoxanthin may have different positions in the light-harvesting complex proteins of the antenna, which provide different levels of energy transfer as a function of light quality (*Premvardhan et al., 2008*). Through pigment analyses, *Brunet et al. (2014)* showed that spectral composition strongly influences the balance between light harvesting and photoprotective capacity of diatoms. *Valle et al. (2014)* and *Schellenberger Costa et al. (2013a)* observed that the energy transfer efficiency of light-harvesting pigments is wavelength-dependent and that diatoms' ability to activate

photoprotection and repair a photodamaged photosystem II (PSII) effectively depends on light quality. *Orefice et al. (2016)* observed that variations in the light spectrum change the photophysiology and biochemistry of diatom cells. Many other wavelength-dependent responses of cyanobacteria and eukaryotic phytoplankton have been observed, especially in laboratory studies of cultures (*Schreiber & Klughammer, 2013*; *Szabó et al., 2014a*, *2014b*; *Herbstová et al., 2015*; *Lawrenz & Richardson, 2017*; *Luimstra et al., 2018*, *2020*). Most field studies of wavelength-dependent acclimation focused on relationships between accessory pigments, the shape of phytoplankton absorption spectra and the underwater light climate (*Hickman et al., 2009*; *Barlow et al., 2017*), but few measured photosynthetic parameters at different wavelengths. Some early studies used the carbon absorption technique and determined α (in multispectral incubators), whose spectral correction through the water column and/or between different water masses has been studied intensively (*Lewis et al., 1985*; *Lewis, Warnock & Platt, 1985*; *Kyewalyanga, Platt & Sathyendranath, 1992*, *1997*; *Kyewalyanga, Sathyendranath & Platt, 2002*).

In the present study, we focused on wavelength-dependent parameters: α, $ETR_{max}$, $E_k$, non-photochemical quenching (NPQ) and high light absorption capacity from 440–625 nm for different natural phytoplankton communities sampled across environmental gradients of a coastal sea. A specifically dedicated device—the multiple excitation wavelength chlorophyll fluorescence analyzer (MULTI-COLOR-PAM) (Heinz Walz, Germany)—was used in its full capacity for the first time in a field study. α, $ETR_{max}$, $E_k$ and $E_{op}$ were determined from PE measurements at five wavelengths as a function of the functional absorption cross section of PSII and NPQ for 19 locations sampled across the English Channel (EC). The EC is an epicontinental sea, particularly suitable for studying photoacclimation strategies of microalgae. This ecosystem has many environmental gradients between coastal and offshore water due to freshwater runoff and high tidal currents (*Brylinski et al., 1991*). This area is dominated by diatoms and the Haptophyceae *Phaeocystis globosa* during the spring bloom (*Houliez et al., 2013a*). Since (i) the wavelength dependence of light absorption capacity is related to the pigment composition of PSII antenna, and (ii) this composition changes in natural samples depending on phytoplankton community structure and specific photoacclimation processes in a given light climate, we tested the hypothesis that phytoplankton $PET_\lambda$ change in shape and level along environmental gradients of light quantity and/or quality, and phytoplankton community structure. The ecological implications of wavelength dependence and plasticity of $PET_\lambda$ parameters are then discussed in the context of turbulent coastal ecosystems. To address these issues, an original analytical approach was developed that used three sequential statistical analyses: linear mixed-effects models, partial triadic analysis and redundancy analysis.

## MATERIALS & METHODS

### Sampling area and strategy

Data were collected during a combined sampling campaign of the JERICO-NEXT program and the 2018 ECOPEL cruise in the EC, from the Strait of Dover (50°58.7′ N, 1°36.64′ E) to Brest (48°20.59′ N, 5°25.03′ W) from 18 April to 2 May 2018 (Fig. 1). Water was sampled

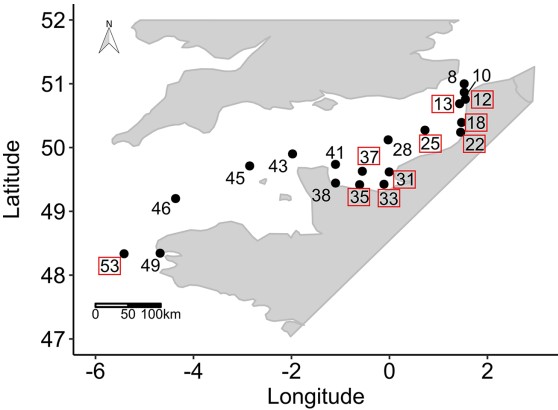

**Figure 1 Sampling locations.** Sampling locations (*n* = 19 among 55 locations sampled) for the study of phytoplankton wavelength dependence in the English Channel during the ECOPEL campaign in April 2018. The framed station numbers refer to thermal and haline stratified water columns.

at 19 locations at a depth of 2 m from inshore to offshore water (using a 20 L Niskin bottle) at different times of day. This zigzag sampling strategy was chosen to consider hydrobiological gradients between coastal/offshore and east/west waters (*Vantrepotte et al., 2007*). To characterize the sampling hour, which can influence phytoplankton physiology, we calculated the number of hours that had elapsed since sunrise (*i.e.* time since sunrise (TSS) in h). The EC is an epicontinental macrotidal temperate system with strong hydrodynamics and substantial river inputs, which provide contrasting light climates that are useful for testing the wavelength-dependence hypothesis of photosynthesis in natural phytoplankton communities. The water bodies sampled were thus used to experiment with different light climates along environmental gradients and with changes in phytoplankton community structure.

## Controlling variables of photosynthesis

Two types of variables that could control photosynthesis were analyzed as *ex-situ* experimental conditions under which the communities grew: (i) abiotic variables and (ii) biotic variables that describe the phytoplankton community structure.

### Abiotic variables: hydrological and light measurements

At each location, conductivity-temperature-depth (CTD) casts were conducted using a SBE25 CTD (Sea-Bird Scientific, USA). Water samples were filtered through a microfiber filter (Whatman GF/C or GF/F), and aliquots were then stored at −20 °C until further processing for dissolved inorganic nutrient concentrations. Concentrations of dissolved inorganic nitrogen (DIN *i.e.* $NO_3^- + NO_2^-$), phosphate ($PO_4^{3-}$), and silicate ($Si(OH)_4$) were measured with a SEAL analytical AutoAnalyzer 3 according to the method of *Aminot & Kérouel (2004)*.

Underwater spectra were measured with a spectroradiometer RAMSES ACC-VIS hyperspectral radiometer (TriOS GmbH, Germany) throughout the euphotic layer, but the present study considered only spectroradiometer measurements from the surface to 2.5 m

depth. Spectroradiometer measurements were made in triplicate (on all spectrum) in the water column, every 50 cm, from the depth where the sensor was not uncovered by the waves. The photon fluence rate was measured every 3 nm from 400–700 nm. Three spectrum bands of interest for photosynthesis (*i.e.* blue (B), green (G) and red (R)) were obtained *via* quantum integration of spectral bands (410–490, 480–580 and 600–700 nm, respectively). These spectral bands, expressed in µmol quanta $m^{-2}$ $s^{-1}$, were chosen according to the study of *Brunet et al. (2014)*. Intensity ratios for three pairs of spectral bands were then calculated (*i.e.* R/B, G/B and G/R) and used as light-quality ratios that depended on the overall chemical and biological characteristics of the waters sampled (*Jaubert et al., 2017*). Finally, averages of these ratios were calculated for replicates of the same depth and over a depth interval ranging between the first depth where it was possible to perform a measurement and the depth of 2.5 m. Depth profiles of photosynthetically active radiation (PAR; 400–700 nm) were obtained using a PAR quantameter (LI-193 4pi from LICOR, USA) connected to the CTD. Vertical diffuse attenuation coefficients for PAR were calculated as follows:

$$K_{d(PAR)} = [\ln(I_0) - \ln(I_z)]/Z \tag{1}$$

where $K_{d(PAR)}$ ($m^{-1}$) is the downwelling diffuse attenuation coefficient of underwater light, and $I_0$ and $I_z$ are photon fluence rates (µmol quanta $m^{-2}$ $s^{-1}$) at the surface and depth z (m), respectively (*Kirk, 2011*).

The depth of the euphotic layer ($Z_{eu}$ in m) was then calculated (Eq. (2)) for each location according to *Kirk (2011)*:

$$Z_{e}u = 4.6/K_{d(PAR)} \tag{2}$$

Vertically averaged light intensity ($E_{avg}$ in µmol quanta $m^{-2}$ $s^{-1}$) in the mixed layer was calculated (Eq. (3)) according to *Riley (1957)*:

$$E_{avg} = I_0 \cdot [1 - e^{(-Kd(PAR) \times Zumixl)}]/(K_{d(PAR)} \times Z_{umixl}) \tag{3}$$

The depth of the upper mixed layer ($Z_{umixl}$ in m) was defined from the CTD profiles using vertical density gradients, caused by vertical temperature and salinity gradients, according to *van Leeuwen et al. (2015)*. The water column was considered to be stratified if the difference in density between the surface layer (0–1.5 m below the surface) and the bottom layer exceeded 0.086 kg·$m^{-3}$ following (*Lowe et al., 2009*). Thus, $Z_{umixl}$ is the depth of the water column without stratification. This approach allowed us to consider that water columns in the EC, which are usually considered to be mixed, may be occasionally stratified and thus influence phytoplankton physiology (*van Leeuwen et al., 2015*). Finally, we calculated the ratio $Z_{eu}/Z_{umixl}$ for each sampling location as a measure of light availability in water, one of the key factors in phytoplankton photoacclimation (*Jensen et al., 1994*).

### Biotic variables: phytoplankton groups, biomass and sample preparation

The FluoroProbe sensor (a multi-wavelength fluorometer, bbe Moldaenke GmbH, Germany) was used to estimate the composition of natural phytoplankton communities, as

in several other studies (*Houliez et al., 2013a*, *2013b*, *2015*). The FluoroProbe distinguished four groups of microalgae *in vivo* and instantaneously: diatoms plus dinoflagellates (*i.e.* "brown microalgae"), Haptophyceae (*Phaeocystis globosa* in the eastern EC (*Houliez et al., 2012*)), Cryptophyceae and Cyanophyceae. The biomass of each group was estimated as an equivalent concentration of Chla ($\mu g\ L^{-1}$). See *Beutler et al. (2002)* for more details about the FluoroProbe. For all photosynthetic parameters, phytoplankton samples were concentrated using a nylon phytoplankton net with a 20 $\mu m$ mesh and 30 cm diameter (Aquatic Research Instrument, Hope, ID, USA), and then kept in the dark under temperature-controlled conditions close to the water sampled and in air-conditioned laboratory conditions before measuring photosynthesis. To measure photosynthesis accurately, the phytoplankton were concentrated to ensure that all samples had the same range of Chla concentration (ca. 100 $\mu g\ L^{-1}$). Phytoplankton biomass in each group was estimated with the FluoroProbe before and after concentrating it.

## Wavelength-dependent photosynthesis parameters and functional absorption cross section of PSII

Wavelength-dependent $PET_\lambda$ was studied using the MULTI-COLOR-PAM, which is particularly suitable for studying the $PET_\lambda$ of phytoplankton (*Schreiber, Klughammer & Kolbowski, 2012*). It provides pulse-modulated measuring light, continuous actinic light, single-turnover light pulses and multiple-turnover or saturation pulses with peak wavelengths at 440 (bright blue), 480 (light blue), 540 (green), 590 (amber) and 625 nm (red light). See *Schreiber, Klughammer & Kolbowski (2012)* for a full description.

Before measuring $PET_\lambda$, samples were first dark-acclimated for 2.5 h (a compromise between the analyses and sampling strategy), without far red exposure (that would have locked the device for too long with respect to the many measurements required). The time of dark acclimation aims to optimize the maximum quantum yield of PSII measurements and neutralize the recent light history of cells (sampled in water columns of different depths and optical properties). It has been shown that there is not an universal protocol and the time required can exceed the classically considered time of 30 minutes and, in certain circumstances, durations of more than 2 hours are necessary (*From et al., 2014*). First, each dark acclimated sample was homogenized within an optical quartz cuvette with a magnetic stirrer then the light sensor US-SQS/WB Spherical Micro Quantum Sensor (Heinz Walz, Germany) was placed into the center of the cuvette to measure the photon flux density at each wavelength. This step provided the "PAR-list" file for each sample, which was used for all later measurements of that sample. Water samples filtered at 0.2 $\mu m$ were used to determine the zero offset (*i.e.* the background signal to subtract from the total fluorescence signal at each wavelength). Next, a subsample of each dark acclimated sample was placed in a 2.5 mL cuvette with a 1 cm path length to adjust the measuring light and gain settings to it in order to obtain the same current fluorescence (Ft) level of $0.5 \pm 0.05$ (relative units) for all wavelengths and to get a good signal-noise ratio. This last step was used to compare fluorescence-rise kinetics (*Szabó et al., 2014a*).

Then, fast kinetic photosynthesis was measured to determine the wavelength-dependent functional absorption cross section of PSII (*i.e.* Sigma(II)$_\lambda$) by measuring O-I$_1$

fluorescence-rise kinetics repeatedly, as described by *Schreiber, Klughammer & Kolbowski (2012)*. Sigma(II)$_\lambda$ was estimated using the pre-programmed fast kinetic trigger file "Sigma1000.FTM", in the same way as *Szabó et al. (2014a, 2014b)* and *Schreiber & Klughammer (2013)* did. In this phase of fast fluorescence, "O" was minimal fluorescence yield corresponding to all PSII reaction centers open. The full closure of PSII reaction centers τ (*i.e.* that of light-driven Q$_A$ reduction during the O-I$_1$ rise) was obtained during a standard one ms long actinic illumination. Sigma(II)$_\lambda$ was calculated according to *Schreiber, Klughammer & Kolbowski (2012)* as:

$$\text{Sigma(II)}_\lambda = 1/(\tau \times L \times PAR) \tag{4}$$

where τ is the time constant (expressed here in seconds) of light driven Q$_A$ reduction determined from the fast fluorescence kinetics measurements, L the Avogadro's constant (6.022.10$^{23}$mol$^{-1}$), and PAR is the quantum flux density (that must be expressed here in mol quanta m$^{-2}$s$^{-1}$) of the light driving the O–I$_1$ fluorescence rise.

Sigma(II)$_\lambda$ was calculated by the user software interface (PAM-Win-3, Heinz Walz) based on the fitted value of the time constant τ obtained from three consecutive measurements separated by 10 s dark intervals, according to (*Klughammer & Schreiber, 2015*). Following this method, the estimate of Sigma(II)$_\lambda$ is independent of Chla concentration. Sigma(II)$_\lambda$ was determined from six subsamples to estimate the mean and variance of each natural community accurately (Supplementary Material, Fig. S1).

Next, automated rapid light curves (RLC) of the PET$_\lambda$ were determined in triplicate (*i.e.* three independent samples) at each of the five wavelengths. For each RLC, samples were exposed to 14 actinic increasing light intensity levels, each 20 s long, as defined in the PAR-list file for each wavelength and sample. Hereafter, "PAR" refers to the photon flux measured at each wavelength, and the same wavelength was always used for the measuring light and actinic light. Saturation pulse settings were defined at a width of 300 μs. The effective quantum yield of PSII (Y(II)) was calculated at each step (Eq. 5). An initial step at 0 μmol quanta m$^{-2}$ s$^{-1}$ was used to determine F$_v$/F$_m$ for samples acclimated to the dark for a long period (2.5 h) (Eq. 6). The relative electron transport rate (r.ETR) was then determined using Y(II), the PAR intensity of the corresponding wavelength (from the PAR-list file) and an arbitrary factor of 0.5 to indicate that PSI and PSII absorb light equally (Eq. (7)). The wavelength-dependent absolute electron transport rate of PSII (ETR(II)) reported in electrons (PSII s)$^{-1}$ was then calculated from Sigma(II)$_\lambda$ (nm$^{-2}$), the Avogadro's constant (L, 6.022.10$^{23}$ mol$^{-1}$) and PAR(II) which is the rate of quantum absorption in PS II, in units of quanta (PS II s)$^{-1}$ according to *Schreiber, Klughammer & Kolbowski (2012)* at each of the five wavelengths (Eqs. (8) and (9)).

$$Y(II) = (F'_m - F)/F'_m \tag{5}$$

$$F_v/F_m = (F_m - F_0)/F_m \tag{6}$$

$$r.ETR = Y(II) \times PAR \times 0.5 \tag{7}$$

$$PAR(II) = Sigma(II) \times L \times PAR \tag{8}$$

$$ETR(II) = PAR(II) \times [Y(II)/F_v/F_m] \tag{9}$$

NPQ (*Bilger & Björkman, 1990*) was calculated as the normalized Stern-Volmer quenching coefficient (Eq. (10)), according to *Lavaud (2007)*:

$$NPQ = F_m/F'_m - 1 \tag{10}$$

All photosynthetic parameters were obtained in triplicate for the five wavelengths of MULTI-COLOR-PAM for each sample.

The *Eilers & Peeters (1988)* model was used to fit r.ETR *vs.* PAR and ETR(II) *vs.* PAR(II) curves to estimate three photosynthetic parameters for each wavelength: light-use efficiency (*i.e.* $\alpha$, the initial slope of the ETR *vs.* PAR curve), the maximum electron transport rate ($ETR_{max}$) and the optimum light parameter ($E_{op}$) in relative (r) and absolute (II) units. The light saturation parameter ($E_k$) was also calculated in the two units as $E_k = ETR_{max}/\alpha$ (*Talling, 1957*). To estimate the degree of photoacclimation of the phytoplankton communities, the $E_{k,440}/E_{avg}$ ratio in $Z_{umixl}$ at sampling was calculated for each location. The $E_k$ ratios (in relative and absolute units) of three pairs of wavelengths (625/440, 540/440 and 540/625 nm) were calculated in the same way as the three pairs of *in situ* spectral bands (*i.e.* R/B, G/B and G/R measured in a water layer of 2.5 m in surface waters).

The Michaelis–Menten model was used to fit NPQ *vs.* PAR curves. A linear regression that forced the intercept to zero was used when the kinetics of these curves differed from the Michaelis–Menten model. Since two models were used, NPQ values were back-calculated using the calibrated models at two irradiances (of PE curves) for each wavelength: low PAR (300 µmol quanta m$^{-2}$ s$^{-1}$) and, for saturating conditions, high PAR (1,200 µmol quanta m$^{-2}$ s$^{-1}$), according to *Szabó et al. (2014a)*.

All PE curves were fitted using the "fitEP" function of the "phytotools" package of R software *R Core Team (2020)* specifically designed to fit phytoplankton photosynthesis curves using simulated annealing (*Silsbe & Malkin, 2015*). The curves for r.ETR *vs.* PAR, ETR(II) *vs.* PAR(II) and NPQ *vs.* PAR were fitted for the three aggregated replicates, and all photosynthetic parameters were obtained at each of the five wavelengths.

## Statistical analysis

All statistical analyses were performed with R version 3.6.0. For abiotic variables, the expectation-maximization with bootstrapping algorithm of Amelia II (*Honaker, King & Blackwell, 2011*) was used to determine missing values ($n = 2$) of light quality data. Principal component analysis (PCA) (*Legendre & Legendre, 2012*) of abiotic and biotic variables was performed using the "PCA" function of the "FactoMineR" package (*Lê, Josse & Husson, 2007*) to determine the internal structure of locations that best explained the variance in each datasets. All data were centered and reduced before performing the PCA analysis.

Statistical analysis of photosynthetic parameters followed a three-step approach (Fig. 2). First, wavelength dependence of each parameter was analyzed using a linear mixed-effects

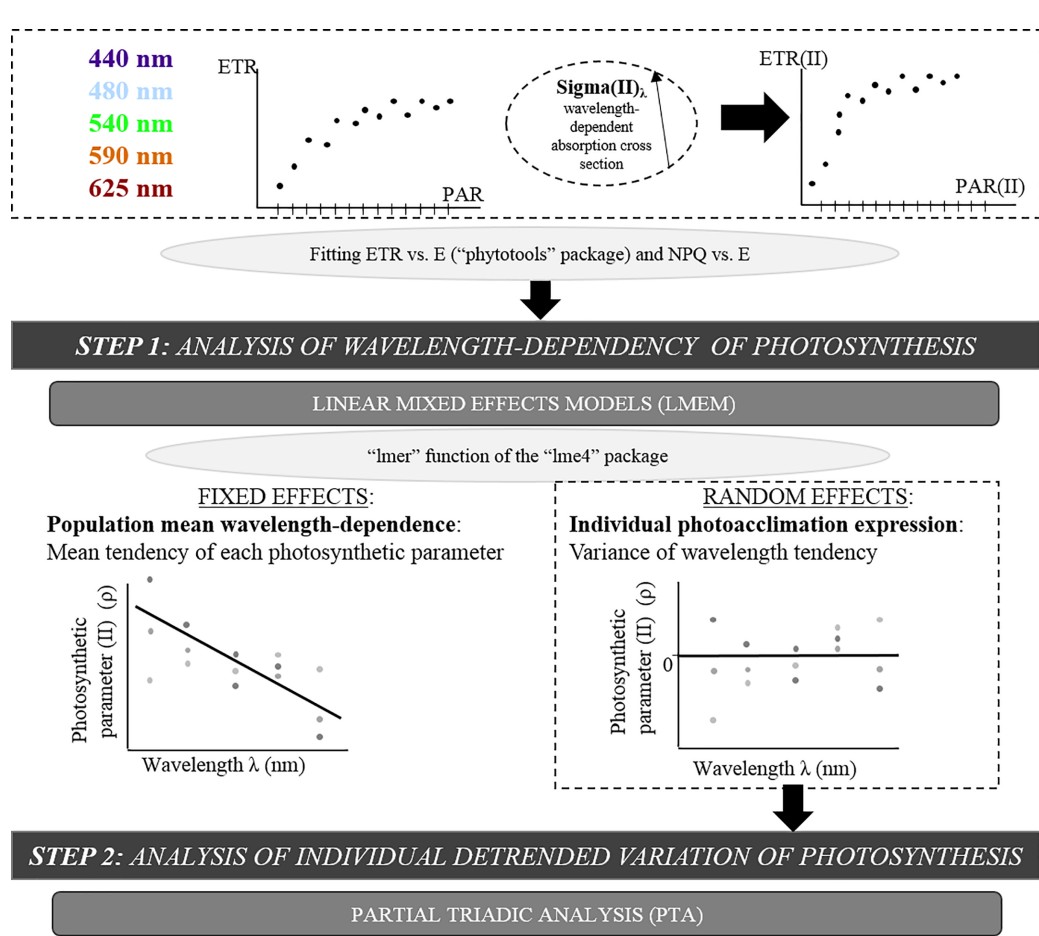

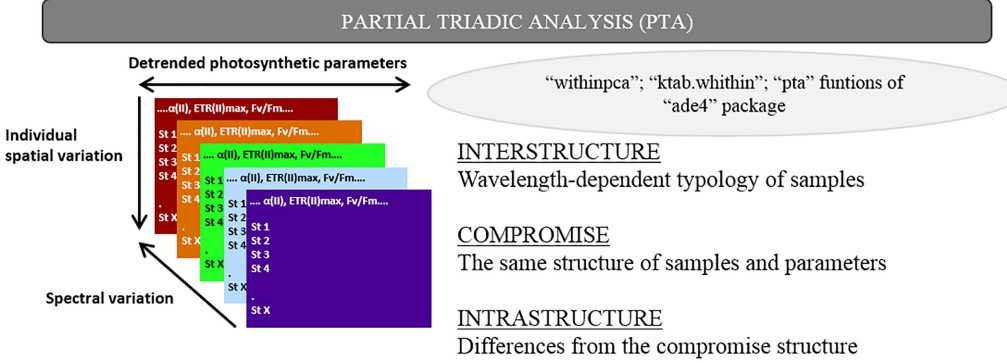

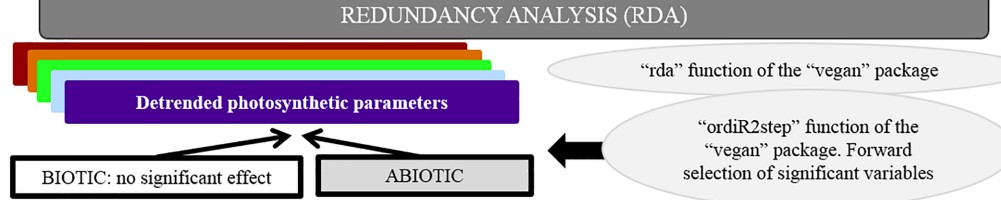

**Figure 2 Original methodological approach diagram.** Diagram of the three-step numerical method used after fitting the electron transport rate (ETR) *vs.* photosynthetically active radiation (PAR) curve: linear mixed-effects models, partial triadic analysis and redundancy analysis.

model (LMEM). Mixed-Effects Models are commonly used to fit regressions to repeated (*i.e.* longitudinal) measures (over time and/or space) by separating the variance explained by the main effects from that explained by random sampling, while considering the wavelength dependence of individuals. The most parsimonious model was linear, and higher-degree polynomials were not significant. LMEMs (*Bates et al., 2015*) were thus used to analyze the population trend across wavelengths for each parameter. LMEMs were fitted using the "lmer" function of the "lmer4" package (*Bates et al., 2015*). Random effects, defined as differences of the locations from the population trend (intercepts and slopes), were used to study individual photoacclimation processes. Wavelengths from 440–625 nm were transposed to 0–185 nm to decrease uncertainty in the model intercept. Hypothesis tests were based on t-tests (for the intercept and slope of fixed effects) and likelihood-ratio tests based on the $\chi^2$ null hypothesis (for random effects) (*Pinheiro & Bates, 2000*).

Second, each wavelength-dependent photosynthetic parameter was detrended by calculating individual differences from the population trend (from LMEMs) and then used in partial triadic analysis (PTA) (*Thioulouse & Chessel, 1987*). PTA analyzes several two-way tables simultaneously (*i.e.* K-tables method). Five tables (one per wavelength) that contained eight photosynthetic parameters (in columns) and 18 locations (in rows) were analyzed (location no. 8 was not considered on PTA analysis due to missing values at 590 nm). Before analysis, all parameter values were centered and reduced based on their overall ranges from all tables. PTA identifies structures that are the same in all tables and assesses their stability among wavelengths. PTA was performed using the "pta" function of the "ade4" package (*Dray & Dufour, 2007*), and related graphics were created with the "adegraphics" package (*Siberchicot et al., 2017*). PTA was applied in three steps— interstructure, compromise and intrastructure analysis—which correspond to co-variance, mean and variance structure analysis, respectively (*Lavit et al., 1994*; *Mendes et al., 2010*).

Third, redundancy analysis (RDA) of the same five wavelength-detrended tables as for the PTA was performed to test for relationships between wavelength-dependent photosynthetic parameters and explanatory abiotic and biotic variables. Data were centered and reduced before analysis. Explanatory variables were selected for the model using an automatic stepwise model (the "ordiR2step" function of the "vegan" package (*Oksanen et al., 2019*)) that performs forward selection based solely on the adjusted $R^2$ and *p*-value (199 permutations). At each step, the variable with the highest additional fit was added to the model.

## RESULTS

### Abiotic and biotic variables

The experimental conditions determined by the abiotic PCA showed contrasting results. For abiotic variables, the first two axes of the PCA explained 66.1% of total inertia (49.5% and 16.6%, respectively) (Fig. 3A). The first axis distinguished samples based on their light-quality ratios (R/B, G/B and G/R), vertical light attenuation coefficient ($K_{d(PAR)}$), $Z_{eu}$, $Z_{umixl}$, salinity and $PO_4^{3-}$ concentration (Fig. 3A). The second axis distinguished samples based on their DIN concentration, temperature, $Si(OH)_4$ concentration and light intensity
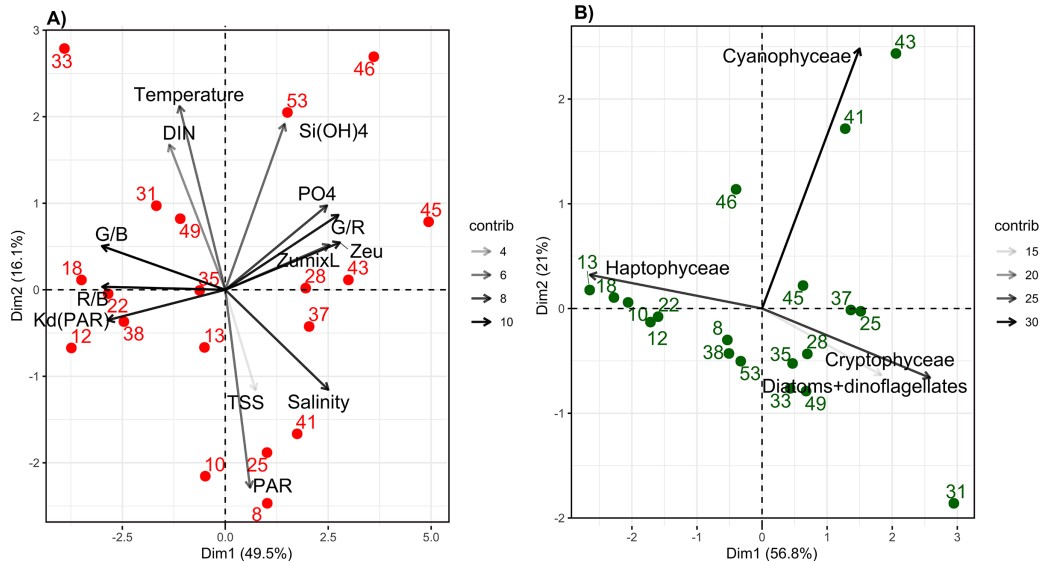

**Figure 3 Principal component analysis (PCA) of abiotic and biotic variables.** The first two axes of the Principal Component Analyses (PCA) performed on: (A) abiotic variables and (B) biotic variables (biomass of the four phytoplankton groups determined by the bbe fluoroprobe), considering the 19 sampling stations. The % of explained variance for each axes is specified. The contribution (contrib) of each variable is indicated by a gray color scaling.

at a depth of 2 m ($PAR_{2m}$) (Fig. 3A). Three groups of samples were distinguished. Group 1 (samples 12, 18, 22, 31, 33, 38 and 49) had intermediate-to-high DIN (>2 μmol L$^{-1}$, up to 30 μmol L$^{-1}$ near Seine Bay) and temperatures (9.5–12.0 °C), low-to-intermediate salinity (<34 PSU) and $PAR_{2m}$ (20–182 μmol quanta m$^{-2}$ s$^{-1}$), the highest R/B and G/B ratios (mean of 0.9 and 1.9, respectively), the highest $K_{d(PAR)}$ (0.2–0.5 m$^{-1}$), the shallowest $Z_{eu}$ (9–23 m) and $Z_{umixl}$ (6.5–21.0 m), and the lowest G/R ratio (±2). On the opposite side of the factorial map, group 2 (samples 28, 37, 43, 45, 46 and 53) had the highest $PO_4^{3-}$ and $Si(OH)_4$ concentrations (>1 and 0.5–2.0 μmol L$^{-1}$, respectively) and salinity (mean of 35 PSU), the lowest R/B and G/B ratios, the highest G/R ratios, the lowest $K_{d(PAR)}$ (0.07–0.17 m$^{-1}$), and the deepest $Z_{eu}$ (26–63 m) and $Z_{umixl}$ (17–78 m). Group 3 (samples 8, 10, 13, 25, 35 and 41) had intermediate salinity (33.–35.0 PSU), the lowest temperatures (<10 °C) and DIN concentration, intermediate light-quality ratios, and the highest $PAR_{2m}$, but with high variability (116–930 μmol quanta m$^{-2}$·s$^{-1}$). Thus, PAR in the abiotic PCA did not distinguish sampling locations well, nor did TSS. Samples had TSS less than 2 h (samples 10, 18, 28, 33, 38, 46, 49 and 53), greater than 10 h (samples 22, 31, 37 and 45) or values between the two (samples 8, 10, 13, 25, 35, 41 and 43). Group 1 had locations near the coast, while group 2 had locations offshore. Detailed information on abiotic variables is shown in the Supplementary Material, Figs. S2 and S3.

The biotic PCA based on FluoroProbe measurements of the biomass of main phytoplankton groups (Fig. 3B) distinguished samples mainly based on the biomass of *P. globosa*; however, the range of variation was low (±25%). The first axis distinguished *P. globosa* from brown microalgae and cryptophytes, while the second axis distinguished Cyanobacteria. The results indicate that *P. globosa* co-dominated with diatoms at several

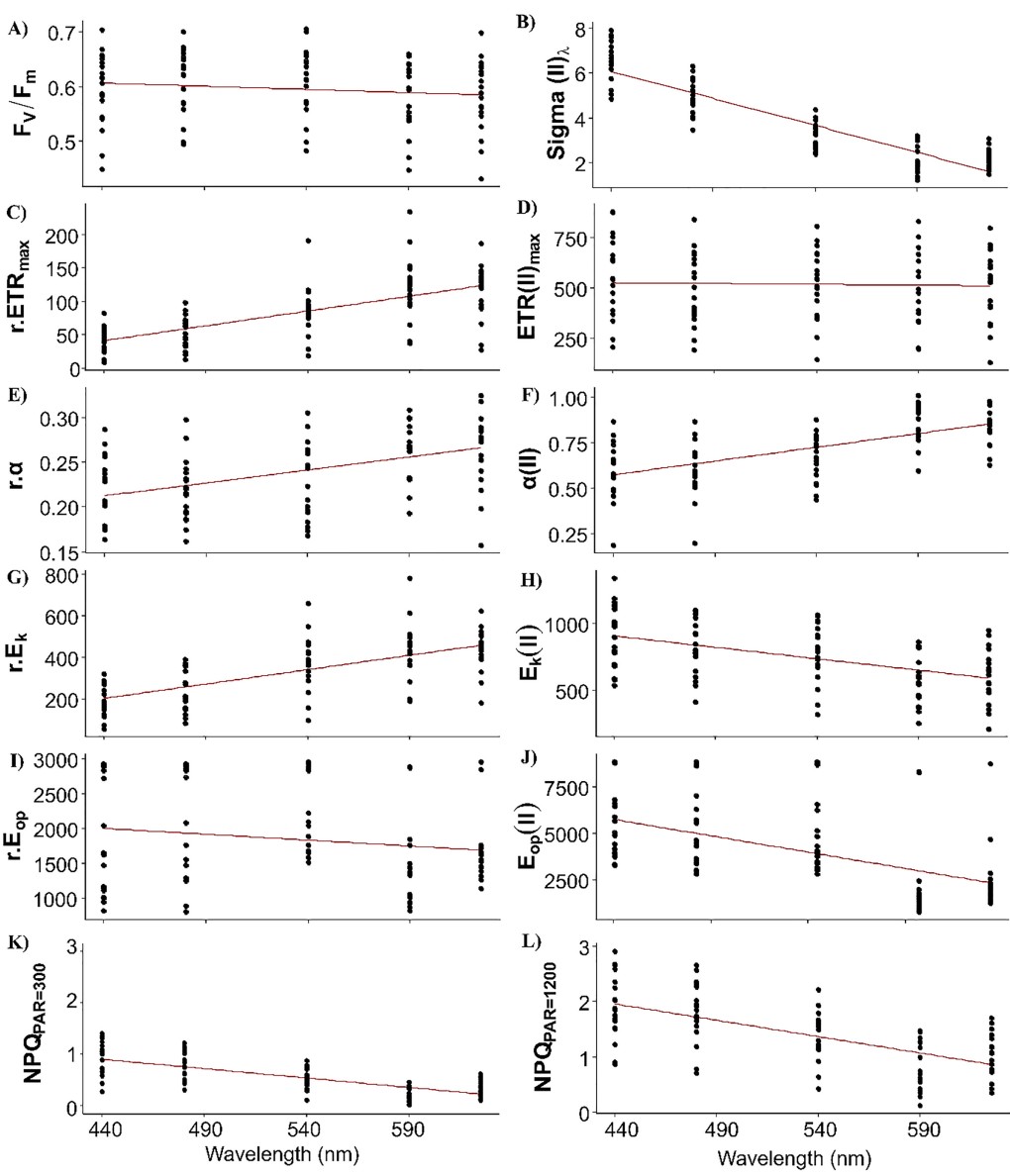

**Figure 4 Raw data of each photosynthetic parameters.** Raw data of each photosynthetic parameter (see Table 1 for definitions) of the 19 samples by wavelength and the fixed effect (red line) of linear mixed-effect models (*i.e.* wavelength dependence at the population level).

locations, and *P. globosa* dominated only samples from locations 10, 12, 13, 18 and 22. Samples from three locations (41, 43 and 46) contained the most cyanobacteria. Details of phytoplankton groups by location are shown in the Supplementary Material, Fig. S4.

## Wavelength-dependent photosynthetic parameters from Linear Mixed-Effects Models

Fixed effects of the LMEMs represented the population trend of each photosynthetic parameter once the spatial nature of the data sampling was considered (Fig. 4).

**Table 2 Statistical outputs of the linear mixed-effects models.**

| | Fixed effects | | | | Random effects | |
| --- | --- | --- | --- | --- | --- | --- |
| Parameters | Intercept | Std. Errors | Slope | Std. Error | Intercept | Slope |
| $F_V/F_m$ | 0.61*** | 0.01 | −0.00012*** | 0.00002 | *** | ns |
| $Sigma(II)_\lambda$ | 6.08 *** | 0.03 | −0.0241*** | 0.0011 | ** | ns |
| $r.ETR_{max}$ | 41.19*** | 5.26 | 0.44*** | 0.04 | *** | *** |
| $ETR_{max}(II)$ | 527.42*** | 35.84 | −0.09 ns | 0.20 | *** | * |
| $r.\alpha$ | 0.21*** | 0.007 | 0.00029*** | 0.00004 | *** | ns |
| $\alpha(II)$ | 0.57*** | 0.03 | 0.0015*** | 0.0001 | *** | ns |
| $r.E_k$ | 200.94*** | 21.78 | 1.39*** | 0.12 | *** | ** |
| $E_k(II)$ | 905.92*** | 43.81 | −1.72*** | 0.32 | *** | ** |
| $r.E_{op}$ | 1,998.90*** | 141.01 | −1.69ns | 0.90 | *** | ns |
| $E_{op}(II)$ | 5,737.18*** | 405.36 | −18.43*** | 2.44 | *** | ns |
| $NPQ_{300}$ | 0.90*** | 0.06 | −0.004*** | 0.0003 | *** | ns |
| $NPQ_{1200}$ | 1.94*** | 0.111 | −0.006*** | 0.0004 | *** | ns |

**Notes:**
Results of the linear mixed-effects models for photosynthetic parameters (see Table 1 for definitions): best estimates of standard error and significance (for fixed effects) and the significance of individual variation (for random effects). Hypothesis tests were based on t-tests (for the intercept and slope of fixed effects), likelihood-ratio tests and the *p*-value based on χ2 statistics (for random effects). Significance codes: ***$p < 0.00$, **$p < 0.01$, *$p < 0.05$, ns: non-significant.

All intercepts were significant, indicating that all parameters differed from zero in the bright blue wavelength (440 nm) (Table 2). Wavelength dependence led to a significant slope of the fixed effect for all parameters except $ETR_{max}(II)$ and $E_{op}(II)$, but the sign of each slope varied among parameters. While the slope of $F_v/F_m$ was significant, its small decrease across wavelengths was considered null for simplicity (Fig. 4A).

$Sigma(II)_\lambda$ is a key parameter since it connects relative and absolute parameters such as $\alpha$, $ETR_{max}$, $E_k$ and $E_{op}$. Population trend of $Sigma(II)_\lambda$ decreased by a factor of 3 across wavelengths (from 6 to 2 $nm^2$, Fig. 4B). Conversely, $r.ETR_{max}$ trend increased by a factor of three (from 50 to 150, Fig. 5C), which may have counteracted the decrease in $Sigma(II)_\lambda$ trend and led to the null slope of $ETR_{max}(II)$ trend (Fig. 4D). Trends of $r.\alpha$ and $\alpha(II)$ increased as wavelength increased (Figs. 4E and 4F), and sharply for the latter, which increased by a factor of two. Since $E_k$ is the ratio of $ETR_{max}$ to $\alpha$, $r.E_k$ increased and $E_k(II)$ decreased as wavelength increased (Figs. 4G and 4H). Since $E_{op}$ values were highly scattered, they were not considered in later analyses (Figs. 4I and 4J). The decreasing trend in NPQs was higher at 1,200 than at 300 µmol quanta $m^{-2}$ $s^{-1}$ (Figs. 4K and 4L). Since NPQ was estimated at low and high PAR from the PE curves, it is interesting to note that the NPQ trends were opposite to those in $r.ETR_{max}$ and $r.\alpha$ (Figs. 4C and 4E respectively), meaning that NPQs could have more influence on the values of these parameters under the blue wavelengths ($NPQ_{440}$ and $NPQ_{480}$ ranged from 1–2) than under the light red wavelength ($NPQ_{625}$ reached 0.2 at PAR = 300 µmol quanta $m^{-2}$ $s^{-1}$). The increasing trend in $\alpha(II)$ across spectrum could thus correspond to a strong decrease in $\alpha(II)$ under the blue wavelengths and not to an optimization under the light red wavelength.

For the random effects, intercepts of all parameters were highly significant, but their slopes were not, except for $ETR_{max}$ and $E_k$ in relative (r) and absolute (II) values (Table 2).

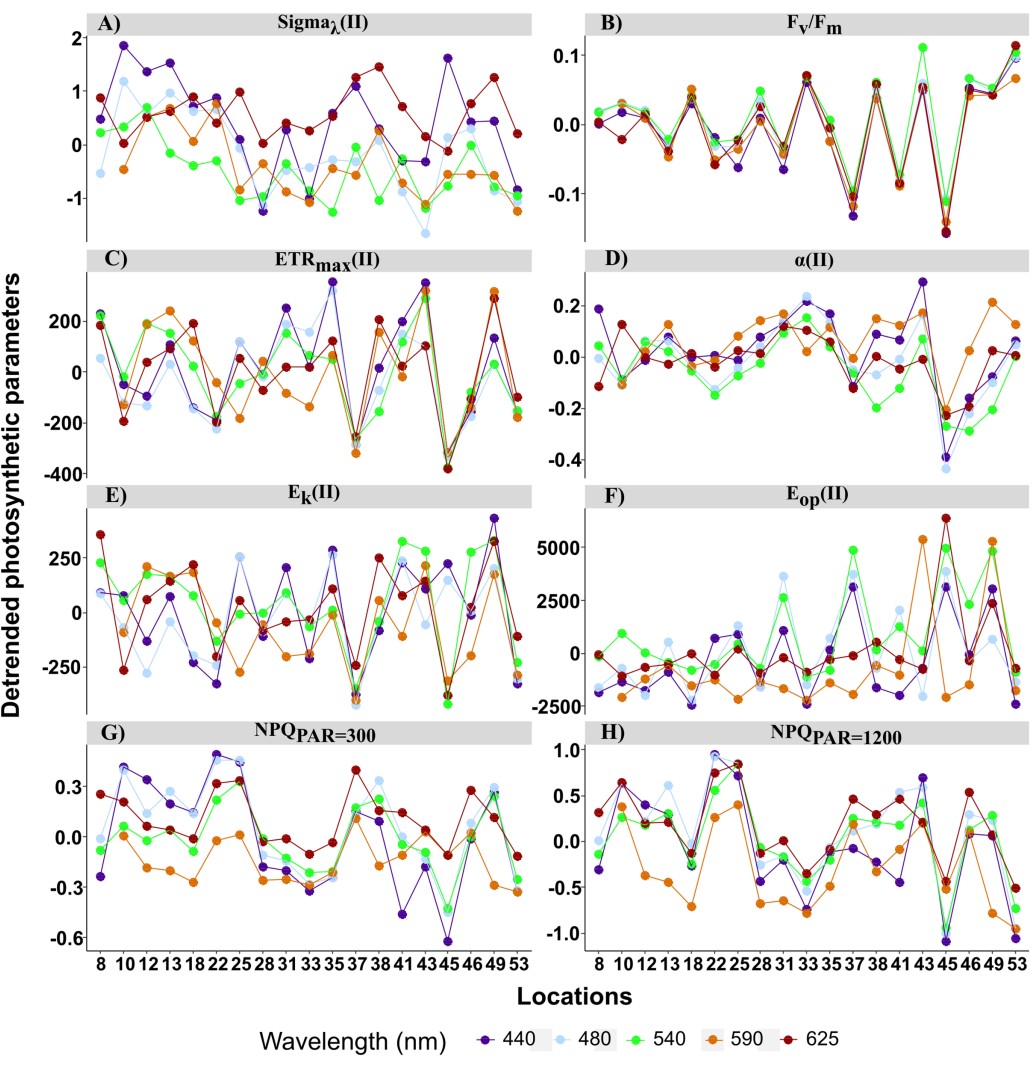

**Figure 5 Detrended spectral photosynthetic parameters.** Detrended spectral photosynthetic parameters (Table 1 for definitions) resulted by calculating individual differences from the population trend (the linear mixed-effect models) among sampling locations. Point colors indicate each of the five wavelengths analyzed: 440, 480, 50, 590 and 625 nm.  

Thus, parameter values differed among samples but, except for $ETR_{max}$ and $E_k$, had the same trend across wavelengths. This resulted in spatial differences in the spectral balance between bright blue and light red wavelengths for $ETR_{max}$ and the photoacclimation parameters $E_k$ (in relative and absolute units for the both).

   When examining detrended values of absolute parameters among locations (Fig. 5), those of $ETR_{max}(II)$ and $E_k(II)$ tended to differ among the five wavelengths by sampling location (Figs. 5C and 5E), unlike those of the other parameters, which were generally more similar among the five wavelengths by sampling location. This was especially true for detrended values of $\alpha(II)$ and $F_v/F_m$, which differed little and almost not at all, respectively, among the five wavelengths by location (Figs. 5A and 5B). Because values of Sigma(II)$_\lambda$ under the light red wavelength were not always the lowest across wavelengths (*i.e.* a slightly

**Table 3 Statistical outputs of each K-tables in partial triadic analysis (PTA).**

| Wavelength | 440 | 480 | 540 | 590 | 625 | Weight | $\cos^2$ |
|---|---|---|---|---|---|---|---|
| **440** | 1.00 | 0.88 | 0.74 | 0.50 | 0.59 | 0.47 | 0.91 |
| **480** | | 1.00 | 0.77 | 0.43 | 0.60 | 0.47 | 0.90 |
| **540** | | | 1.00 | 0.54 | 0.69 | 0.47 | 0.89 |
| **590** | | | | 1.00 | 0.61 | 0.38 | 0.69 |
| **624** | | | | | 1.00 | 0.44 | 0.80 |

Note:
Vector correlation coefficients between the submatrix of photosynthetic parameters at each wavelength (nm), their weights in partial triadic analysis and $\cos^2$.

non-linear distribution) (Fig. 4B), its detrended values under the light red wavelength were higher than the population trend (Fig. 5A). According to the statistical analyses, however, a linear model fit best to Sigma(II)$\lambda$ values.

## Sample wavelength dependence of samples from the PTA

The first two axes of the PTA interstructure explained 85.38% of total inertia (Supplementary Material, Fig. S5), and the five wavelength tables had similar weights (0.38–0.47; Table 3) and a significant representation ($\cos^2$ close to 1; Table 3). The PTA was thus adequate overall and highlighted similarities among the wavelengths. All five wavelengths were positively correlated and positively projected on the first axis (ca. 71.24% of the total inertia; Supplementary Material, Fig. S5). The second axis separated the bright blue and green wavelengths from the amber and light red wavelengths. The amber wavelength differed the most from the others and had the same correlation with the first and second axes.

When projecting the wavelength-dependent photosynthetic parameters on the compromise coordinates it explained 66.49% of total inertia (Table 4, Fig. 6A), the relative positions of polygons indicated that PTA results generally met our expectations: opposition between the group of $F_v/F_m$, $\alpha$(II) and $ETR_{max}$(II) *vs.* the group of Sigma(II)$_\lambda$ and $NPQ_{300-1,200}$, with parameters related to photoacclimation ($E_k$ and $E_{op}$) between these two groups. We observed the well-known relationships between variables related to energy flows ($F_v/F_m$ and $NPQ_{300-1,200}$), related to Sigma(II)$_\lambda$, and the major parameters that control PE relations in absolute units—$\alpha$(II) and $ETR_{max}$(II)—which had a positive overall correlation.

The intrastructure of the PTA showed differences between photosynthetic parameter patterns among the five wavelengths (Figs. 6A to 6E). Patterns for the main parameters, such as $ETR_{max}$(II), $\alpha$(II), Sigma(II)$_\lambda$ and $NPQ_{300-1,200}$, changed from the bright blue wavelength (440 nm) to light red wavelength (625 nm). The most evident change was the rotation of $\alpha$(II) and NPQ300–1,200 respected to the spectral pattern of Sigma(II)$_\lambda$. At blue wavelengths Sigma(II)$_\lambda$ and $NPQ_{300-1,200}$ were inversely correlated to $\alpha$(II) (Fig. 6A) but at amber and light red wavelengths there were any correlation (Figs. 6D and 6E). Sigma(II)$_\lambda$ was incorrectly represented in the main plane at 625 nm (Fig. 6E). The PTA intrastructure analysis also showed patterns for the locations among the five

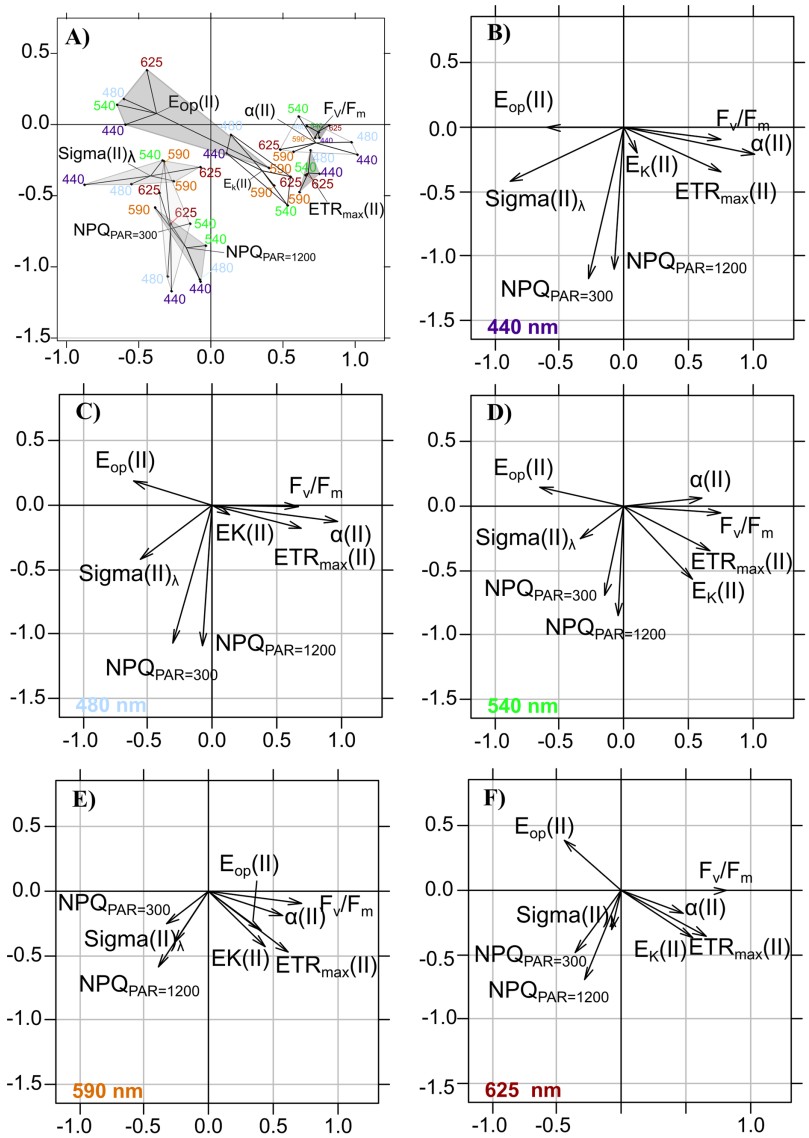

**Figure 6 Infrastructure results of detrended photosynthetic parameters of Partial Triadic Analysis.** Results of intrastructure of the partial triadic analysis of detrended photosynthetic parameters ETR-max(II), Ek, Alpha(II), $E_{op}$(II), Sigma(II)$_\lambda$, Fv/Fm and NPQ (see Table 1 for definitions at 300 and 1,200 µmol quanta m$^{-2}$s$^{-1}$) projected on compromise coordinates (A). Each photosynthetic parameters coordinate is represented separately for each wavelength at 440 (B), 480 nm (C), 540 (D), 590 (E) and 625 (F) nm. See Table 3 for compromise contribution to total inertia and Table 4 for weight and cos$^2$ of each of the wavelengths colors in PTA analysis.

wavelengths (Fig. 7). Wavelength dependence differed greatly among locations: polygons were largest for locations 33, 35, 43, 37 and 45, and smallest for locations 13, 18, 31, 46 and 53 (Fig. 7). In addition, the blue wavelengths (440 and 480 nm) displayed a general circular change among locations, moving from the right of the polygon for location 53 to the left for location 45 (Fig. 7).

**Table 4 Total inertia of partial triadic analysis (PTA) compromise analyses.**

| Axe | Inertia | Cum | Cum (%) |
|---|---|---|---|
| 1 | 10.56 | 10.56 | 37.76 |
| 2 | 8.54 | 19.10 | 66.49 |
| 3 | 4.51 | 23.61 | 82.18 |
| 4 | 2.42 | 26.03 | 90.61 |

Note:
Total inertia of partial triadic analysis (PTA) compromise analyses, cumulative inertia of each PTA axis (Cum) and percentage of cumulative total inertia (Cum %).

**Table 5 Results of redundancy analysis (RDA) of detrended photosynthetic parameters at each wavelength.**

| Wavelength | Model | Variance | Residual variance | Abiotic | | Biotic | |
|---|---|---|---|---|---|---|---|
| | | | | adj $R^2$ | $p$ | adj $R^2$ | $p$ |
| 440 | $Z_{eu}$ | 1.60 | 6.40 | 0.15 | ** | 0.023 | ns |
| 480 | $Z_{eu}$+TSS+$K_{d(PAR)}$+DIN | 4.16 | 3.82 | 0.38 | *** | −0.001 | ns |
| 540 | $Z_{eu}$ | 2.35 | 5.65 | 0.20 | ** | 0.010 | ns |
| 590 | TSS+G/R | 2.57 | 5.42 | 0.23 | ** | 0.05 | ns |
| 625 | None | 6.00 | 1.99 | 0.10 | ns | −0.111 | ns |

Notes:
Forward-selected explanatory variables of RDA model, explained and residual variances, the adjusted $R^2$ and associated $p$-value for abiotic ($Z_{eu}$ (euphotic layer), TSS (Time since sunrise), $K_{d(PAR)}$ (downwelling diffuse attenuation coefficient of underwater light), DIN (of dissolved inorganic nitrogen, *i.e.* $NO_3$+$NO_2$) and G/R (Green/Red light quality ratio)) and biotic variables. No biotic variable was selected, and the model for the red wavelength (625 nm) was not significant. Significance codes: ***$p$ < 0.00, **$p$ < 0.01, ns: non-significant.

## Explanatory variables of wavelength dependence from RDA and linear regression

The RDA results showed that abiotic variables related to light emerged first as explanatory variables (Table 5). Euphotic depth ($Z_{eu}$) was selected the most often, for three of the five wavelengths (Table 5), from 440–540 nm. TSS, which represents the recent light history of cells, was selected twice, for the light blue wavelength (480 nm) and the amber wavelength (590 nm). The G/R light ratios were selected for the amber wavelength (590 nm). DIN concentration was the only non-light parameter selected, for the light blue wavelength. No variables were selected for the light red wavelength (625 nm), and $PAR_{2m}$ did not seem to influence the parameters.

To further explore the influence of $Z_{eu}$ and the difference in control of photosynthetic parameters under bright blue and light red wavelengths, we sought specific connections between the photoacclimation parameter $E_k$ and $Z_{eu}$ (in ratios with $E_{avg}$ and $Z_{umixl}$, respectively), and between the ratio of $E_k$ (in relative and absolute units) measured at 625 and 440 nm and the corresponding R/B light ratio ($E_{625/440}$) in water masses. Two significant linear trends were found between the $E_{k,440}/E_{avg}$ ratio (in relative and absolute units) and the $Z_{eu}/Z_{umixl}$ ratio for stratified water columns (Fig. 8A for graphs and correlation coefficients). Correlations were non-significant for non-stratified water columns for r.$E_{k,440}$ and $E_k(II)_{440}$, as well as under the other wavelengths. Considering all

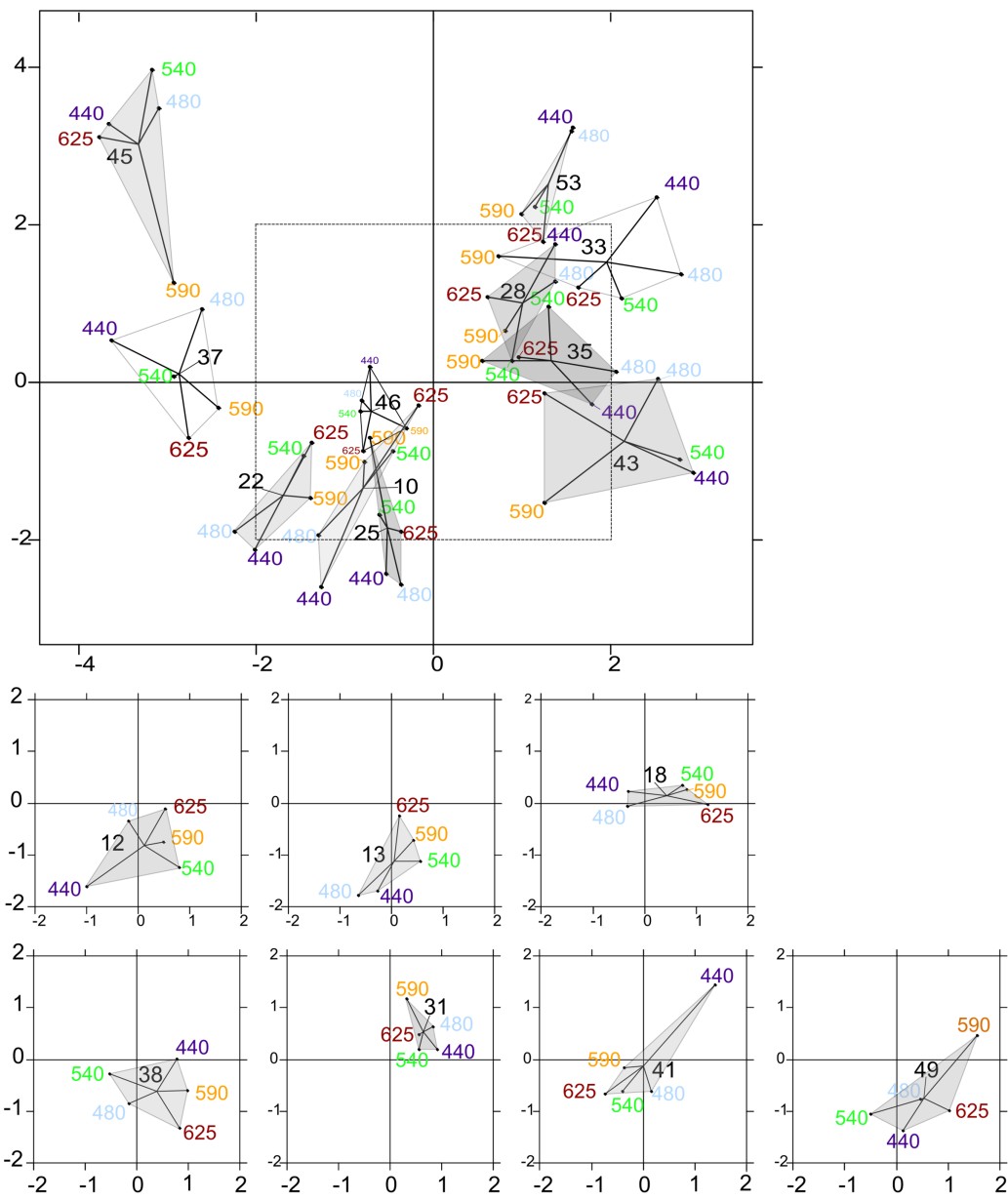

**Figure 7 Intrastructure results of partial triadic analysis (PTA) for each location.** Partial triadic analysis intrastructure (first and second dimension) projected on compromise coordinates of photosynthetic parameters for each of the 18 locations: factorial map of spectral responses projected on compromise coordinates for each sample. The dotted line in the first figure represents the coordinates used to represent samples 12, 13, 18, 38, 41 and 49 separately to improve visualization of this part of the factorial map.

sampling locations revealed other significant correlations between the absolute ratio $E_k(II)_{625/440}$ and (1) the red/blue light ratios in surface waters $E_{625/440}$ (see Fig. 8B for correlation coefficients) and (2) the TSS factor (r = −0.65, n = 19, p < 0.05, Supplementary Material, Fig. S6).

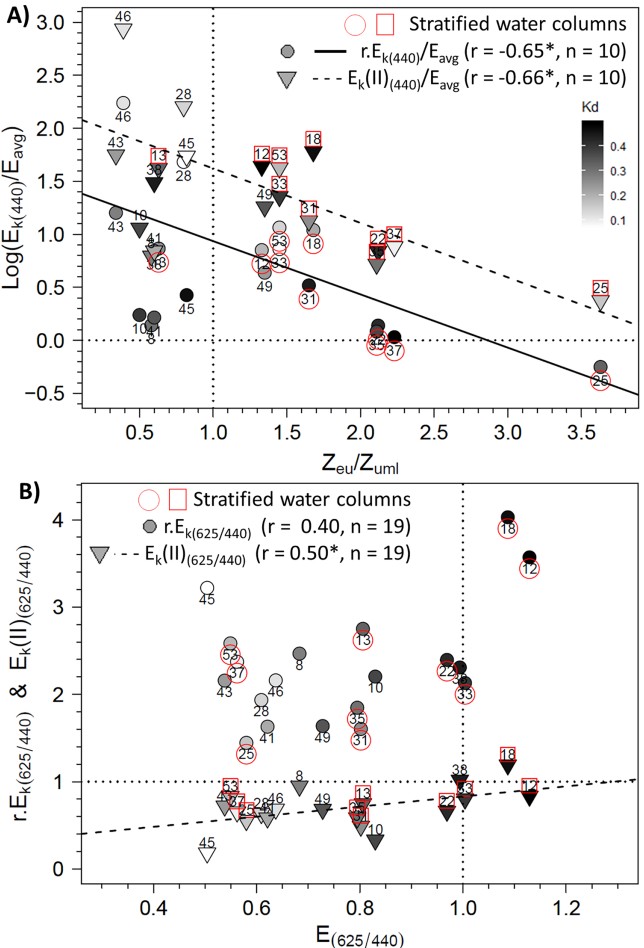

**Figure 8 Relationships between photoacclimation indexes and physical parameters.** Relationships between (A) the $E_{k,440}/E_{avg}$ ratio (the photoacclimation index measured at 440 nm to the vertically averaged PAR light intensity) (in relative (r) and absolute (II) units) and the $Z_{eu}/Z_{uml}$ ratio (depth of the euphotic layer to that of the upper mixed layer) and (B) between $E_k(II)_{(625/440)}$ (ratio of photoacclimation index measured at red and blue wavelengths) and the corresponding red/blue wavelength ratios of light ($E_{(625/440)}$) in water masses for the 19 locations. The regression equation in B is y = 0.5789 x + 0.2516 (F = 5.62; $p = 0.0298$). Pearson correlation coefficients, level of significance (*$p < 0.05$) and the number of considered data are also reported on each graph.

## DISCUSSION

Our results provide new insights into the wavelength dependence of photosynthetic parameters and PSII functional absorption cross section of coastal water phytoplankton communities. To date, the literature has focused on one or two species under a few growth conditions in the laboratory (*Schreiber & Klughammer, 2013*; *Brunet et al., 2014*; *Szabó et al., 2014a*; *Luimstra et al., 2018*) or older *in situ* studies that focused on the wavelength dependence of α in the water column (*Lewis, Warnock & Platt, 1985*; *Lewis, Ulloa & Platt, 1988*; *Kyewalyangaa, Platt & Sathyendranath, 1992*). Studying the present dataset including all photosynthetic parameters was complex due to environmental gradients and changes in community structure but made possible by the use of three powerful statistical methods. The wavelength dependence of photosynthetic parameters that was characterized

at the population level and the sample level, will be discussed first in relation to the theories on photosynthesis and photoacclimation, and then from an ecological point of view.

## Physiological meaning of wavelength dependence of photosynthetic parameters

### Light absorption capacity

Since light absorption was measured according to the method of *Schreiber, Klughammer & Kolbowski (2012)*, Sigma(II)$_\lambda$ can be considered an intrinsic property of the PSII units for each sample. Thus, the recent light history at sampling did not change the light absorption capacity of the 19 samples studied. Cell light absorption was thus a function of only pigment composition of photosynthetic units (*Schreiber, Klughammer & Kolbowski, 2012*), even though the packaging effect may slightly skew the relation between Sigma(II)$_\lambda$ measurements and pigment concentration (*Gorbunov et al., 2020*). Since the slopes of the fixed (population level) and random (individual level) effects were significant and non-significant, respectively, all of the phytoplankton communities absorbed more light in the blue spectral range and with the same wavelength dependence, regardless of the sample.

Given the sampling area, we expected the decreasing population trend in Sigma(II)$_\lambda$ across wavelengths. This is a typical result for cell communities dominated by brown microalgae such as diatoms with Chla as the main light-absorbing pigment (*Kuczynska, Jemiola-Rzeminska & Strzalka, 2015*). The small increase under the light red wavelength (625 nm) is also consistent with this result (since Chla also absorbs red light), and with other Sigma(II)$_\lambda$ measurements for diatoms using the same method (*Goessling et al., 2018b*). This is consistent with our group-based community-structure measurements, which show that the diatom-dinoflagellate and haptophyte groups dominated all 19 samples. Consequently, the phytoplankton communities may not have been composed of species with completely opposite strategies for light absorption, as is common in experimental and theoretical modeling studies (*Luimstra et al., 2019*; *Burson et al., 2019*). Thus, the following discussion focuses on the plasticity of typical brown microalgae communities to the wavelength dependence of light climates in coastal water depending on their physiological state.

Conversely, the absence of significant individual wavelength dependence for Sigma(II)$_\lambda$ among the 19 samples was unexpected; however, the light absorption capacity differed among samples regardless of the wavelength. Signature changes over the spectrum were expected because the spectral quality of light is crucial for microalgae to achieve optimal photoacclimation in the face of variable light quality and intensity during growth (*Valle et al., 2014*). Light quality can influence gene expression that adjusts pigment and protein compositions of specific antenna complexes (*e.g.* fucoxanthin-chlorophyll a/c-binding antenna pigment-protein complex of diatoms called FCP), which changes spectral absorption by cells. Conversely, light quantity can influence only pigment concentrations (*Valle et al., 2014*). In the present study, the changes in light quality may have been too small to induce significant changes in spectral absorption capacity among the 19 samples due to hydrodynamic conditions and the daylight cycle, which agrees with the results of *Gorbunov et al. (2020)*. Cells could have managed variations in incident light quality

*via* their photosynthetic apparatus without having to change their pigment composition. However, changes in light intensity were sufficient to produce significantly different spectral absorption capacities among samples.

The relatively high light absorption capacity of cells under the green wavelength (540 nm) agrees with results of *Goessling et al. (2018b)* for a planktonic diatom and for microphytobenthic diatoms (*Goessling et al., 2018a*), but does not agree with those of a brown-gold microalga *Nannocloropsis oculata* (Eustigmqtophyceae) (*Szabó et al., 2014a*) using the same method as in the present study. This could be because some or all cells in the communities of the present study were acclimated to green wavelengths in the water column. The green light ratios (G/B and G/R) measured always exceeded one, indicating that green wavelengths dominated in all sampling conditions. It is well known that green light dominates coastal water during the algal blooms because the high chlorophyll concentration absorbs blue wavelengths (*Kirk, 2011*). This absorption capacity in the green spectral band could be due to an increase in the "green" absorption capacity of fucoxanthin, which absorbs more energy from 390–580 nm due to the basic structural and functional properties of fucoxanthin and its location in the light-harvesting complex proteins of the antenna (*Premvardhan et al., 2008*; *Kuczynska, Jemiola-Rzeminska & Strzalka, 2015*). *Burson et al. (2018)* also showed continuous light absorption spectra, with better green light absorption capacity for a fucoxanthin diatom than for green and blue-green phytoplankton, in agreement with *Valle et al. (2014)*.

### General physiological state

The population trend across wavelengths and $F_v/F_m$ values indicate that cells were in a good physiological state, considering the community composition based on diatoms and *Phaeocystis globosa*, regardless of the wavelength and thus were not under high nutrient stress. The population trend of $F_v/F_m$ was centered on 0.6, and $F_v/F_m$ values were never less than 0.4. Generally, the theoretical maximum of $F_v/F_m$ is 0.7 (in a dark-adapted state, as in this study), and its critical value is 0.3 (*Painter et al., 2010*) for communities dominated by brown microalgae (diatoms and dinoflagellates). We will not discuss here the complex links between nutrients and photosynthetic parameters, this is not our topic and this was done specifically in other works for the same ecosystem (*Napoléon, Raimbault & Claquin, 2013*). In addition, the good nutrient status of water is confirmed by comparing nutrient concentration measurements in the present study to the seasonal nutrient cycle established for the EC (*Gentilhomme & Lizon, 1998*). Using a similar measurement method, *Szabó et al. (2014b)* observed similar values of $F_v/F_m$ across wavelengths (which approached 0.7) for cultures in nutrient-replete conditions. *Gorai et al. (2014)* observed spectral independence of $F_v/F_m$ when comparing physiological properties of a culture under blue and white lights.

### Spectral trends and photoacclimation processes

In this context of samples with the same wavelength dependence of light absorption and good physiological state, wavelength photoacclimation can be investigated through trends in $ETR_{max}(II)$, $\alpha(II)$, NPQ and $E_k$. Although $r.ETR_{max}$ and $r.\alpha$ had a significant trend

across wavelengths, they do not provide enough information to understand the spectral dependence of photosynthesis in detail, which is a function of light absorption, given here by Sigma(II)$_\lambda$ (*Schreiber, Klughammer & Kolbowski, 2012*). These relative parameters are useful only to compare and better understand the influence of Sigma(II)$_\lambda$ on spectral photosynthetic processes in absolute units. Both parameters had a positive slope across wavelengths, indicating that r.ETR$_{max}$ and r.$\alpha$ spectra were the inverse of the Sigma(II)$_\lambda$ spectrum; conversely, in absolute units, the slope of ETR$_{max}$(II) disappeared, while the positive slope of $\alpha$(II) was maintained. These results were unexpected since water was sampled at different times of the day, and cells were kept in the dark for 2.5 h before being analyzed. An explanation could be that cells in the samples were in a wavelength-dependent photoprotective state due to the underwater light climate.

We expected the trend of ETR$_{max}$(II) to decrease over the spectrum because the cells grew in a natural environment in which blue and green light dominated, since red light is quickly absorbed in the water column (the R/B ratios were usually less than one). Several studies involving phytoplankton and microphytobenthos observed increasing values of ETR$_{max}$(II) in the blue wavelength (*Mercado et al., 2004*; *Szabó et al., 2014a*; *Goessling, Cartaxana & Kühl, 2016*; *Goessling et al., 2018a*) related to changing light conditions. Opposite results have also been found. *Schreiber, Klughammer & Kolbowski (2012)* indicated that photoinhibition could explain the decrease in ETR$_{max}$(II) under blue light, that the time needed to recover the starting values is much longer under bright blue than light red wavelengths, and that the recovery under blue light remains only partial after several hours. *Correa-Reyes et al. (2001)* observed that growth rates of eight microphytobenthic species decreased more under blue light than light of other colors. In the present study, given the higher light absorption in blue (440 and 480 nm) than light red wavelengths (625 nm), and since the high energy of blue wavelengths can cause photodamage (*Dougher & Bugbee, 2001*), the lack of slope for ETR$_{max}$(II) likely reflects a photoinhibition or photoprotective state of the communities towards blue wavelengths, which initially decreased r.ETR$_{max}$ values under blue light. For ETR$_{max}$(II), significant wavelength dependence among the 19 samples is superimposed on the non-significant population trend. Thus, some samples had different degrees of photoinhibition/ photoprotection under blue light independent of other light colors. We can thus speculate about wavelength dependence of photoinhibition/photoprotection mechanisms.

We expected high r.$\alpha$ and especially $\alpha$(II) under blue wavelengths and thus a decreasing trend from 440–590 nm, with a small increase at 625 nm. The variation in $\alpha$ as a function of light quality is the best known photosynthetic parameter because it was closely studied from the mid-1980s to the early 2000s by carbon-14 incorporation, for calculations of primary production rate (*Lewis et al., 1985*; *Lewis, Warnock & Platt, 1985*; *Lewis, Ulloa & Platt, 1988*; *Kyewalyangaa, Platt & Sathyendranath, 1992*; *1997*; *Kyewalyanga, Sathyendranath & Platt, 2002*). At worst, a null slope of $\alpha$(II) over the spectrum, like for ETR$_{max}$(II), was expected, but the consistently lower values in the blue wavelengths were unexpected given the Sigma(II)$_\lambda$ variations observed and previous studies of natural communities (*Lewis et al., 1985*) or brown microalgae cultures of a red tide dinoflagellate (*Schofield, Prezelin & Johnsen (1996)* talking about Pyrrophyta). This result does not reflect

an improvement in red light-use efficiency, since brown microalgae requires 24–48 h to photoacclimate to changes in red light (*Valle et al. (2014)* about the marine diatom *Phaeodactylum tricornutum*), which is consistent with properties of the underwater light climate. Unlike red light, high-intensity blue light can cause rapid changes in cells, such as energy allocation between photosynthetic and photoprotective pathways in coastal species (*Lavaud, 2007*; *Brunet & Lavaud, 2010*). As many experimental studies show, the cycle of xanthophyll (a photoprotective pigment) and NPQ are fundamental photoprotective processes that are activated within seconds to minutes to dissipate excess absorbed light energy (*Dimier et al., 2009*). Energy dissipation by carotenoids can reduce photosynthetic rates under blue light, which increases NPQ (*Brunet et al., 2014*). Many other studies have used NPQ to measure the overall photoprotective capacity of the photosynthetic apparatus (*e.g. Dimier et al., 2007*; *Lavaud, 2007*). The xanthophyll cycle photoprotective mechanism has been observed in the coastal sea of the EC for the same locations the phytoplankton communities in this study have been collected (*Brunet, Brylinski & Lemoine, 1993*; *Brunet & Lizon, 2003*) and in permanently well-mixed ecosystems (*Alderkamp et al., 2011*). Therefore, like for $ETR_{max}(II)$, the decrease in the trend of $\alpha(II)$ observed over the spectrum could be due to photoinhibition and, more likely, photoprotection. Photoinhibition of $\alpha$ has been experimentally verified (*Björkman & Demmig, 1987*; *Baker & Bowyer, 1994*).

The hypothesis of a photoprotective influence on $\alpha$ (in relative and absolute units) requires that photoprotection occur early and at low light intensities. Consistent with this hypothesis, NPQ population trends were significant and values were always higher under blue than red light at low intensities. These results are confirmed by those of *Goessling et al. (2018a)* for suspensions of phytobenthic diatoms and those of *Tamburic et al. (2014)* for a brown-gold microalgae*Nannocloropsis occulata* (Eustigmqtophyceae). In the present study, NPQ was 1 and 2 under low- and high-intensity blue light, respectively. According to *Lefebvre, Mouget & Lavaud (2011)*, NPQ usually exceeds 1 for cells that face the sun and are not well adapted to high-intensity light.

Thus, since communities in the present study had adequate photoprotective capacity against blue light, maximum quantum efficiency $F_v/F_m$ was always high regardless of the wavelength, while effective quantum efficiency, which influences $\alpha$ and $ETR_{max}$, decreased early under blue light (and probably green light). This created slightly different spectral signatures for $\alpha(II)$, which is determined under low light intensity, among the samples depending on the *in situ* light intensity, which thus likely decreased wavelength effects. Conversely, $ETR_{max}(II)$ is determined under higher light intensity, in which individual spectral effects were efficient. To support this hypothesis, blue light is expected to be less prone to cause photoinhibition such as photodamage than red light (*Brunet et al., 2014*), and cells have greater PSII repair capacity under blue light than red light, since photoprotection and PSII repair are induced by protein-encoding genes under blue light, but are lacking under red light (*Valle et al., 2014*).

In this context, $E_k$ varied across wavelengths due to relative variations in $ETR_{max}$ and $\alpha$, since $E_k = ETR_{max}/\alpha$. Values of $r.E_k$ were indeed lower in the blue wavelengths, while those of $E_k(II)$ were higher. These results are fairly consistent with our hypothesis for

communities in the photoprotective state rather than the photoinhibition state, as are the high $F_v/F_m$ values under blue wavelengths. This is supported by the two significant correlations observed between the $E_{k,400}/E_{avg}$ ratio (in relative and absolute units) and the $Z_{eu}/Z_{umixl}$ ratio, indicating that the communities were in good agreement with the high light availability in the mixed layer (*Jensen et al., 1994*; *Wang et al., 2011*). This is particularly true for the $rE_{k,400}/E_{avg}$ ratio, which tended towards the reference value of 1 at four locations that had stratified water columns. Values close to one indicate optimization of absorbed light by photosynthetic metabolism (*Anning et al., 2000*; *Anning, Harris & Geider, 2001*). To our knowledge, these two correlations are new results for an ecosystem known for its high hydrodynamic regime. Most studies in dynamic coastal ecosystems observed cells in a photoacclimation state and $r.E_k$ values less than or greater than one (*Claquin et al., 2010*; *Houliez et al., 2013a*). One exception is *Jouenne et al. (2005)* in the Baie des Veys (French coast), who used carbon-14 measurements rather than active fluorescence measurements. The photoprotective state, based on NPQ under *in situ* irradiance, includes well-known processes that can operate continuously to protect microalgae from potential photoinhibition and, after photodamage, correspond well to photoacclimation processes (*Alderkamp et al., 2013*). The physiological plasticity of phytoplankton in limiting photodamage usually explains much of the diurnal variation in photosynthetic processes (*Schuback et al., 2016*). These include many processes of the photosynthetic apparatus that influence $r.E_k$ and $E_k(II)$ to match the *in situ* light intensity (*Dubinsky & Stambler, 2009*; *Schofield et al., 2013*).

The advantage of measuring photosynthesis at several wavelengths using the functional light absorption capacity of natural phytoplankton is revealed by the differing trends of the photosynthetic parameters observed over the spectrum. Comparing the spectral trends and estimating the absolute photosynthetic parameter were necessary to identify the photoacclimation state of the 19 samples.

## Ecological meaning of sample spectral variability and controlling factors

To investigate the ecological meaning of our results for wavelength dependence, we addressed the variability in photosynthetic parameters outside of the population trends. This approach was based on the detrended measurements of the PTA and RDA by wavelength, and the relationships between the $E_k(II)$ ratios and their controlling factors.

According to the PTA, the covariations observed among detrended values of photosynthetic parameters and their change over the spectrum are consistent with wavelength dependence at the population level of the parameters studied with the LMEMs. The PTA showed that the pattern of the parameters changed gradually and consistently between blue and red wavelengths, concretely for light absorption and photochemical quenching. Using the PTA to scan the intrastructure of the 19 samples revealed also that spectral variation patterns of the parameters of each sample differed from each other in size, shape and position on the factorial map. Sizes and shapes of these spectral polygons were not related to water column stratification (*e.g.* the polygon of the most stratified location (no. 33) was the same size as that of location 43, which was not stratified).

According to the RDA, photosynthetic parameters measured by wavelength were influenced mainly by euphotic depth. The forward selection of the RDA first retained the most correlated variable in a group of abiotic variables that covaried with biotic variables. Consequently, since the euphotic depth ($Z_{eu}$), the light extinction coefficient ($K_{d(PAR)}$) and the three light-quality ratios were collinear or correlated variables (in the abiotic PCA), individual photosynthetic parameters could also be controlled by the underwater light quality and turbidity, since the RDA also selected $K_{d(PAR)}$ and a wavelength ratio. Considering all photosynthetic parameters, the main RDA result is thus consistent with the significant relationship observed between the light saturation ratio ($E_k(II)_{625/440}$) and R/B light ratio ($E_{625/440}$). The RDA also selected the TSS variable discussed later. Mixing depth ($Z_{umixl}$) was another interesting ecological parameter selected by the RDA. The RDA confirms the hydrodynamic regime's control of the photoacclimation index $E_k$ previously displayed by the relationship between $E_k$ (r and II) and $Z_{eu}/Z_{umixl}$. Since physical forcing in a given upper mixed layer controls certainly the level reached by $E_k$, wavelength dependency of photosynthetic parameters is generally a trade-off between the light quality of the different encountered water masses and changes in light quantity throughout the upper mixed layer due to the hydrodynamic regime. The RDA results and singular correlations with $E_k$ (r and II) are consistent, but the correlations of $E_k(II)_{625/440}$ with critical environmental parameters provide better understanding of the ecological mechanisms that influence phytoplankton photoacclimation. The results are interesting because they were obtained from the field, where light-quality ratios (between red and blue wavelengths for instance) are known to change with depth (Supplementary Material, Fig. S7), as also described by *Brunet et al. (2014)* and *Jaubert et al. (2017)*. It is likely that the influence of vertical changes in light-quality ratios on microalgae acclimation was small in the present study, especially because most upper mixed layers were shallow (6.5–14 m) and the residence times of the cells at a given depth were low. The spectral light saturation ratios $E_k(II)_{625/440}$ were therefore not correlated with the $Z_{eu}/Z_{umixl}$ index.

The significant correlations between $E_k(II)_{625/440}$ and $E_{625/440}$ or TSS clearly highlight that natural phytoplankton communities can implement photoacclimation processes that are driven by the *in situ* light quality that also change during the daylight cycle. Most water masses had $E_{625/440}$ ratios less than one, which indicates water in which blue wavelengths dominate red wavelengths, and displayed higher $E_k(II)$ phytoplankton photoacclimation index in blue wavelengths than in red wavelengths. This original result is valid only for absolute $E_k(II)_\lambda$ parameters related to $Sigma(II)_\lambda$, not for relative parameters $r.E_k$. Since $Sigma(II)_\lambda$ measured with the MULTI-COLOR-PAM is an intrinsic property of microalgae (*Schreiber, Klughammer & Kolbowski, 2012*), and since microalgae can increase pigment concentration to absorb more light (*Lawrenz & Richardson, 2017*), differences in pigment concentrations may change the level of $Sigma(II)_\lambda$ measured during the day. Due to the differing variations in $r.E_k$ between blue and red wavelengths, $E_k(II)_{625/440}$ ratios are therefore correlated with $E_{625/440}$ ratios in the water masses and with TSS, the time elapsed since sunrise. These relationships provide new information about the natural environment and are consistent with many experiments under controlled conditions. As several studies of monospecific cultures subjected to contrasting R/B ratios show

(*Schellenberger Costa et al., 2013a*; *Brunet et al., 2014*), variations in the light spectrum and in blue *vs.* red wavelengths influence photoprotective capacity and the pigment composition of phytoplankton. *Schellenberger Costa et al. (2013a)* conclude that photoprotection is regulated more by light quality (especially blue wavelengths) than by the overall light intensity. *Kirk (2011)* stated that phytoplankton detect not so much the spectra of light, but rather differences between wavelength ratios received by PSI and PSII, using blue and red wavelength photoreceptors (*Jaubert et al., 2017*) that regulate photosynthesis and promote photoacclimation (*Schellenberger Costa et al., 2013b*; *Petroutsos et al., 2016*). This explains why the $E_k(II)$ for the green wavelength did not correlate significantly with the *in situ* light-quality ratios of the green wavelengths. Photoacclimation mediation by *in situ* blue wavelengths, as discussed by *Schellenberger Costa et al. (2013a)*, is thus consistent with our field study.

The $E_k(II)_{625/440}$ *vs.* $E_{625/440}$ correlation, like the abiotic PCA, indicates indirectly that variables such as temperature, $PAR_{2m}$, and DIN and $Si(OH)_4$ concentrations do not influence wavelength photoacclimation greatly. Previous studies in the EC showed that abiotic variables were the main variables that controlled spatial and/or temporal variations in relative photosynthetic parameters (*Jouenne et al., 2007*; *Napoléon et al., 2013*; *Houliez et al., 2015*). However, the variables that control these photosynthetic parameters may vary among geographic areas and/or seasons. In the present study, this correlation was determined over a large spatial scale.

## Ecological implications and consequences

It is the general question of the absorption capacity of light in relation to the quality of light and its impact on primary production that is discussed here.

The precise examination of the $E_k(II)_{625/440}$ *versus* $E_{625/440}$ relationship in link with the reference values of 1 indicates that the two ratios matched each other well. For example, at location 38, the $E_k(II)_{625/440}$ and $E_{625/440}$ ratios equaled one, which could be because the water there was sampled before sunrise. However, other communities sampled before sunrise (*e.g.* at locations 28, 46 and 53) showed imbalances in their $E_k(II)_{625/440}$ ratios related to the $E_{625/440}$ ratios of water masses. The regression model indicates that to observe an $E_k(II)_{625/440}$ ratio of 1, a theoretical $E_{625/440}$ ratio of 1.29 would be required (which is close to our measurements). In this case, the blue wavelengths would decrease to 77 μmol quanta $m^{-2}$ $s^{-1}$ given a red light of 100 μmol quanta $m^{-2}$ $s^{-1}$. Thus, the $E_{625/440}$ ratio cited above indicates no strong imbalance in available energy and thus no stressful ecological situation for phytoplankton. In spring in temperate water, blue wavelengths are absorbed due to CDOM, in link with terrestrial discharge near estuaries and/or phytoplankton blooms themselves (*Vantrepotte et al., 2007*; *Astoreca, Rousseau & Lancelot, 2009*). *Lawrenz & Richardson (2017)* studied photoacclimation under extreme conditions, with a total absence of blue light (*i.e.* black water with high CDOM concentrations). They showed that, depending on the taxon, microalgae retain or lose their initial light absorption capacity on the spectrum, and their absorption capacity adapts to the light quality to which they were exposed, even with red wavelengths. Some species can survive under red light in the short term, but in the long term, cytoplasmic structures and

chloroplast membranes degrade (*Humphrey, 1983*) and Chla concentrations decline (*Forster & Dring, 1992*). *Rivkin (1989)* showed the strong influence of blue light on carbon fixation and incorporation into amino acids and proteins. The $E_k(II)_{625/440}$ *vs.* $E_{625/440}$ relationship in the present study cannot be extrapolated beyond the measurement limits (*i.e.* to the completely unbalanced light ratios of *Lawrenz & Richardson (2017)*), but includes representative conditions generally found in the EC or temperate systems.

In comparison, $E_k(II)_{625/440}$ ratios near 0 would indicate that blue wavelengths are ultra-dominant in the water mass and could activate strong photoprotection mechanisms or even cause photodamage, which would decrease $ETR_{max}(II)$ greatly under blue wavelengths. Under these conditions, given the intercept of the linear model, $E_k(II)$ values under blue wavelengths would be only 25% of those under the red wavelength, which suggests that primary production would decrease greatly. However, electron flows under green, amber and light red wavelengths remained high, and NPQ was not as strong as under blue wavelengths. These results were consistent with the low light absorption (Sigma $(II)_\lambda$) under these wavelengths, as was the NPQ. However, this raises the issue of using the spectral approach to calculate primary production based on the photosynthetic parameters of RLC relationships.

Previous studies of the wavelength dependence of photosynthesis specified the systematic error produced by measuring α under white-light incubators when comparing incubation light climates (*Laws et al., 1990*) or primary production models (*Kyewalyanga, Platt & Sathyendranath, 1992*). Most classic incubators do not reproduce light spectra at the low intensities that phytoplankton encounter in the water column because of the high variability in light quality with depth, but also with time, due to vertical mixing in the upper part of the water column. *Schofield, Prezelin & Johnsen (1996)* examined the error caused by using the same or different α from cultures grown under different light qualities when calculating primary production in a theoretical and simplified water column. Depending on the species and growing conditions, differences between vertical primary production rates estimated by the two calculation methods ranged from 12–49%. Other studies showed that α values measured on board under artificial light (*Irwin et al., 1990*) could be corrected from the shape of the phytoplankton absorption spectrum (*Kyewalyanga, Platt & Sathyendranath, 1997*; *Sathyendranath et al., 1999*). This approach involves the field of remote sensing in particular and includes "optical" and "full spectral" models (*Platt & Sathyendranath, 1988*; *Sathyendranath & Platt, 1993*; *Behrenfeld & Falkowski, 1997*). Recent studies (*Kovač et al., 2017*; *Sathyendranath et al., 2020*) combined a spectral model of underwater light with a model of the integrated spectral response of algal photosynthesis consistent with photoacclimation processes. These studies recommend using the photosynthesis action spectrum or spectral correction of α in the water column, especially when differences in the shape of the action spectrum of α are larger than those in its magnitude at each wavelength (*Sathyendranath & Platt, 1993*). However, these studies did not consider that light-quality ratios in surface water can also greatly influence photoacclimation of microalgae. This was revealed in the present study by some individual wavelength-dependence phenomenon that differed significantly among the samples, and the $E_k(II)_{625/440}$ ratios, which involve $ETR_{max}(II)$ and α(II) (*i.e.* the

photosynthetic apparatus), especially the functional absorption cross section of PSII and the maximum rate of $PET_\lambda$, according to the optical definition of $E_k$ (*Falkowski & Raven, 2007*). Parameters $\alpha(II)$ and $Sigma(II)_\lambda$ showed no significant individual wavelength dependence among water masses, unlike $ETR_{max}(II)$ and $E_k(II)$, which are involved in primary production at high light intensities. The central issue is thus how photosynthetic activity induced by wavelengths beyond 480 nm can compensate for the decrease in photosynthesis under strong blue light when estimating primary production in different water masses. Sensitivity analysis of physicochemical properties of water would pave the way for future research on wavelength dependence of phytoplankton photosynthesis, as well as spectral dependence at the seasonal scale, using current active fluorescence measurement technologies.

## CONCLUSION

Our results indicate that natural phytoplankton communities can photoacclimate to light quality dynamically under contrasting environmental conditions in temperate coastal seas in response to the available energy balance between red and blue wavelengths.
The wavelength dependence of photosynthetic parameters was here characterized at the population level (in a consistent way with the photosynthesis theory) and at the sample level where a high spatio-temporal variability was observed. The photosynthetic parameters $E_k$, $ETR_{max}$ (both in relative and absolute units) and NPQ proved to be here the most important ones for understanding the photoacclimation dynamics of natural microalgae communities. With a general model of photoprotection against blue light based on NPQ of photosynthesis, the present study shows that natural phytoplankton communities were most adapted to high-intensity light when a large amount of light was absorbed (*e.g.* blue wavelengths) but appeared "shade" adapted when low-intensity light was absorbed (*e.g.* green, amber and light red wavelengths), to paraphrase *Nielsen & Sakshaug (1993)*.

The results for dynamic photoacclimation processes showed a general trade-off between light quality and intensity, and also all related light factors (*e.g.* $K_{d(PAR)}$, $Z_{eu}$, $Z_{umixl}$), which is difficult to find in experimental studies, through which photoacclimation has been discussed for many years (*Schofield, Prezelin & Johnsen, 1996*; *Brunet et al., 2014*; *Gorai et al., 2014*). Experimental studies are generally performed with monocultures growing in the comfort of laboratories for generations and rarely with natural communities using conventional photosynthetic parameters. These cultures are subjected to photon flux of different wavelengths and/or intensities, with different frequencies of variation, different daylight cycles, etc. Experiments often consider the controlling variables separately but combining them simultaneously seems relevant for understanding processes of photoacclimation to light variations in the field, as discussed by *Combe et al. (2015)* in a modeling study.

## ACKNOWLEDGEMENTS

The authors thank Dr L.F. Artigas that conceived and led the field campaign (ECOPEL-2018), the captains and the crews on board the R/V « ANTEA » for assistance, as well as

the technical staffs of CNRS-LOG. The field campaign was carried out thanks to the convention (n° 2101893310) between the French Ministry for the Ecological and Inclusive Transition and the CNRS for the implementation of the Marine Strategy Framework Directive (MSFD). This study is also integrated into the European project H2020 JERICO-NEXT.

### Funding

This work has been financially supported by the European Union (ERDF), the French State, the French Region Hauts-de-France and Ifremer, in the framework of the project CPER MARCO 2015-2021. M. Michel-Rodriguez benefits from a PhD grant from the French government (Ministère de l'Enseignement Supérieur et de la Recherche/Université de Lille). The funders had no role in study design, data collection and analysis, decision to publish, or preparation of the manuscript.

### Grant Disclosures

The following grant information was disclosed by the authors:
European Union (ERDF): CPER MARCO 2015-2021.
Ministère de l'Enseignement Supérieur et de la Recherche/Université de Lille.

### Competing Interests

The authors declare that they have no competing interests.

### Author Contributions

- Monica Michel-Rodriguez performed the experiments, analyzed the data, prepared figures and/or tables, authored or reviewed drafts of the paper, and approved the final draft.
- Sebastien Lefebvre conceived and designed the experiments, analyzed the data, authored or reviewed drafts of the paper, and approved the final draft.
- Muriel Crouvoisier performed the experiments, authored or reviewed drafts of the paper, and approved the final draft.
- Xavier Mériaux performed the experiments, authored or reviewed drafts of the paper, and approved the final draft.
- Fabrice Lizon conceived and designed the experiments, performed the experiments, analyzed the data, prepared figures and/or tables, authored or reviewed drafts of the paper, and approved the final draft.

### Data Availability

The raw data are available in the Supplemental Files.

### Supplemental Information

Supplemental information for this article can be found online at http://dx.doi.org/10.7717/peerj.12101#supplemental-information.

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
