# Peer review of "Underwater light climate and wavelength dependence of microalgae photosynthetic parameters in a temperate sea"

_PeerJ, doi:10.7717/peerj.12101_

## Round 0.1 · original submission · Minor Revisions

The manuscript requires some improvements before publication. Further details and clarifications are required and reviewers also suggested that rephrasing is necessary about the methodology from a photophysiological point of view. Some parts of the discussion and abstracts also require revision. Please revise the manuscript follow the reviewer's indications below.

Reviewer 1 ·

Basic reporting

The manuscript by Michel-Rodriguez presents a large scale study of the wavelength dependent photosynthetic properties of natural phytoplankton groups, by using wavelength-dependent chlorophyll fluorescence analysis. The manuscript presents a valuable large scale assessment of several natural locations supported by a rigorous statistical analysis. The manuscript is clearly written in professional manner. However, some clarifications, further details and rephrasing is necessary about the methodology of chlorophyll fluorescence analysis and the interpretation of photosynthetic processes and parameters.

Experimental design

The aims and the research questions of the work are clearly defined and the experimental design was carried out with high standards and with proper rationale. Some further improvements are suggested in particular to clarify the details and explanation of the methods, as detailed below.

line 89: multi-colour pulse amplitude modulation fluorometer could be called as ‘fluorometer’, not a sensor, to be more specific.

line 102 and other places – the naming of the colors is somewhat inconsistent, blue is called dark blue at some places, the 590 nm spectral region is amber or orange – please be more consistent with colour labelling

line 167-170: hypothesis is defined, but it is not clear in the context of Fig. 1 and the Figure 1 also needs further clarifications. In each panel, 2 lines are presented, but it is not explained what do these actually show. Why does light quantity and quality response decrease with wavelength? Photosynthetic organisms has a defined absorption spectrum as well as an action spectrum, which depends on the pigment composition, functional light absorption etc of the given species. These properties may of course change with the light climate, due to chromatic adaptation and adaptation to different irradiance levels and the authors may wanted to define the hypothesis accordingly, however this is not clear from Figure 1.

line 252: dark adaptation time should be explained and justified. Is 2.5 h dark adaptation is the right time period to obtain maximal Fv/Fm? In some algal species, very long dark adaptation can lead to the operation of chlororespiration, which might lead to over-reduction of electron transport chain, which eventually cause a decline in Fv/Fm even in darkness. Therefore the authors should give more details on the selected dark adaptation methodology to give a clear reference physiological state for the multi-wavelength Chl fluorescence analysis. Furthermore, it is not clear whether far-red light was used for the multicolour PAM measurements. Far-red light would serve the purpose of keeping the electron transport chain oxidized and thus it would help establishing reproducible control conditions for the measurements (Schreiber et al. 2012). This may or may not be beneficial for the presented work, however it should be clearly described whether such reference state conditions were applied.

line 278: for light curve measurements, the authors presumably applied 14 actinic light levels or steps, not pulses. Please clarify.

line 306: define Zumixl at its first mention.

line 367: the details of PCA should be described in Materials and Methods. Accordingly, in Figure 4 more explanation of the PCA analyis and the denotions and labelings on the X and Y axes should be explained in the figure legend as well.

Validity of the findings

The results presented in the work are properly discussed. All related data is provided with high details and the conclusions are well supported by the findings. Some further improvements are suggested particularly for the Discussion, as detailed below.

line 529: define briefly the fucoxanthin chlorophyll protein antenna complex.

line 538-542: in this study not a green microalga, but Nannochloropsis (Eustigmatophyceae) was studied. Nannochloropsis sp. lacks Chl b and it lacks strong absorption capacity in the 540 nm spectral region, so it shows a quite different Sigma(II) spectrum as compared to the Sigma(II) spectrum of planktonic diatoms. Therefore it is somewhat unclear what do the authors mean that the result 'does not agree with those of a green microalga'. Other studies showed that the spectrum of the absorption cross-section did not change significantly as a result of chromatic adaptation or photoacclimation to various irradiance levels in Nannochloropsis (see e.g. Kandilian et al. Bioresource Technology 137 (2013) 63–73). Therefore the Discussion should be rephrased in this aspect when comparing different microalgae species with respect to their different inherent pigment composition.

line 545-550: the extended absorption of fucoxanthin into the 'green' spectral region is not only due to its specific location and high amount in the light-harvesting antenna, but also as a result of its specific molecular structure, as it contains a carbonyl moiety on a polyene backbone (Premvardhan et al. 2008 and other cited works). As the main phytoplankton group analyzed by the authors are diatoms and other Chl a/c containing organisms, authors should reflect on the basic structural and functional properties of fucoxanthin and FCPs.

line 554-557: ‘The theoretical maximum of Fv/Fm is 0.7 (in a dark-adapted state, as in this study), and its critical value is 0.3’ - for which species is this true? The absolute or 'healthy' level of Fv/Fm can be very different across species, for example in some cyanobacteria the optimal Fv/Fm level is 0.2-0.5, due to the contribution of fluorescence of phycobilisomes. Therefore, when defining a critical threshold of Fv/Fm for ‘healty’ values, it always has to be defined under which conditons and for which species this is valid for.

line 566-568: ‘Although r.ETRmax and r.alpha had a significant trend across wavelengths, they do not provide useful information about the spectral dependence of photosynthesis’. Some clarifications may be necessary here. The photosynthesis vs. irradiance curves, or 'ETR curves' should display wavelength specific features, because if the light absorption is higher at 440 vs 590 nm or 625 nm, alpha should be higher and consequently sub-saturating irradiance, Ek (and ETRmax) should be lower at 440 than at 625 nm (see e.g. Figs 4-5 in Schreiber et al. 2012). Therefore, the 'relative' ETR parameters have important wavelength specific features. When the incident PAR is multiplied with Sigma(II), the PAR(II), the quantum absorption rate can be determined, from which the absolute electron turnover rate of PSII could be determined (eqs. 7 and 8), which theoretically is independent of wavelength (provided that the Sigma(II) scaling is correct).

line 573: ‘These results were unexpected…’ could it be the reason that some of the photosynthetic parameters exhibited diurnal fluctuations? These might show different patterns in different species, therefore it is hard to reconcile its role in a phytoplankton community.

Figure 5: the deviation is high in nearly all parameters, therefore it is hard to see any wavelength dependent trend. Perhaps only Sigma(II) and NPQ shows some trend. Apparently ETR(II) is suitably equalized because of the Sigma(II)-scaled relETR.

·

Basic reporting

The paper is written in a good English style, the figures and tables are okay and literature is well referenced. Sometimes the text might be shortened and results/conclusion should be accentuated, as they are lost in long methodological/statistical descriptive text.

Experimental design

The paper is in the scope of the journal and fill knowledge gap. Some information about light measurements is missing or is yet not clear. Also some of the fluorescence measurements still need additional information.

For further details, see review report

Validity of the findings

The paper shows new findings on community level sampled in the field. It supports knowledge already known on the species level, but extends it to natural communities. The statistical treatment I can’t judge, as I am not an expert on it, but it seems reasonable to me.

Additional comments

Although the results of your study are important for estimating primary production in different water bodies, the outcome is not amazing, as it follows what is already known about photosynthetic adaptations/acclimation. However, these results support the common knowledge on a species level now on the community levels in the field, which will be useful for oceanographic research. So a publication is recommended.
But from a photophysiological point of view, there evolve several questions for me, which need to be answered before publication. There are also some interpretations which to my opinion deviated from the analysis, which look too much mechanistic for me. Also I miss some methodological informations.
To investigate communities consistent of such different algae groups is really a hard work. There are multifactorial effects, including photoprotection, absorption cross section, sieve effects (population density), transmission spectrum, irradiance level, nutrient concentration, sun position, season etc., which need to be considered. For that, interpretation is difficult and has to be done carefully, as outcome of this analysis.
If possible the text should be straightened, and results/conclusion could be more accentuated.

In details, I will go sequentially through the manuscript with questions and comments:

Line 97: most photosynthetic studies are done with red light, also for in vivo fluorescence or gas exchange measurements. Blue LED are a recent technical development and are only recently on the market.
Line 113ff: It is a misbelief that brown algae are more efficient in photosynthesis the red ones. Ecologically, red algae/cyanobacteria are still growing at water depth where irradiances are too low for brown ones, because the construction of their photosynthetic apparatus is more efficient in energy transfer, nearly without energy loss. Moreover, the green enhanced spectrum typically for coastal water is best absorbed by the phycobilines.
Line 123ff: Do not mix low and high light conditions in one sentence. It makes it difficult to follow. Larger antenna size is a result of low light, higher reaction center number with smaller antenna an acclimation to high light conditions.
Line 135: energy absorption of fucoxanthin does not change; however, energy conduction can change with different position in the antenna construction.
Line 169: It would be better to have a clearer indication of the axis in fig 1. The direction in which photosynthesis changes or where wavelengths are longer are not visible.
Line 201: Surface reference measurements are made above or below the water border? It is important because of reflection which depends on the sun position and sensor characteristic (refractive index). If you have considered only spectra close to the water surface, how did you determine the absorption characteristic of the respective water body? You certainly need a greater depth range, to see differences within the respective spectrum. From only PAR measurements this will be not visible. Information is missing.
Line 213: Also here, atmospheric measurement or sensor covered by water?
Line 243: Did you control the phytoplankton diversity by a microscopic check? The FluoroProbe check is not enough, as especially Haptophyceae pigments are similar to those of the Heterokonts. With the low number of detection channels of the instrument you might see changes within a community, but really it is hard to discriminate between the different algae groups with a similar pigment content.
Line 250: 440 nm I would not call dark blue it is nice bright blue and 625nm is light red with a touch of orange. Real red would be seen at 650 nm.
Line 254: ? Suggestion: First, each sample was homogenized within an optical cuvette by a magnetic stirrer and then the light sensor was placed into the center of a quartz cuvette to measure the photon flux density at each wavelength using a US-SQS/WB….
Line 262: Fluorescence level of the dark adapted samples? It is not clear if dark or light adapted sample. You may have adjusted to a similar Fo by the gain or measuring light irradiance, to get a good signal-noise ratio.
Line 278: better: 14 pulses of increasing actinic light intensity, each 20 s long, as defined in the PAR-list file
Line 268: PSI and PSII absorb light equally? This depends on the antenna size of each photosystem! You mean you need at least two photons to drive both photosystem (factor 0.5 for one photosystem). The light absorption/energy conduction of the respective photosystem depends also on state transitions.
Line 375: I miss methodological information how the spectra light ratios were determined (see above) . It would be good to get information, in which extent this has changed at the different sample locations, as this is one important point of this study.
Line 393: from locations 10 to 22 ? (see fig 4B)

General comment to Fig.5 : Biophysically spoken, with constant photon number, red light effect should be twice as high as blue light, as half of the energy of blue photons is lost by heat dissipation (S2->S1 singlet state). Thus, NPQ (heat dissipation) must be higher in blue than red light (Fig.5 K,L). So you might say, red light is more effective. However blue light also activates the sensor needed for photomorphological effects in algae, which use e.g. cryptochoms. However, this is independent from the energy usage of the photosynthetic apparatus. Thus, the higher energy content of blue photons is wasted compared to those of red photons. This shows your analysis, if it is not standardize to the absorption cross section, as the antenna in brown algae absorb more blue than red light. Only in red algae, the higher Chl content of PS I increase the Chl absorption cross section of PS I compared to PS II in the blue and red waveband. Here, PS II absorbs more in the green/orange waveband, which therefore is more effective.
Eop and NPQ is in the orange waveband lowest, as in your brown community it is less absorbed, and less effective.

Line 418: (NPQ625 reached 0.2) at PAR 300.
Chapter 3.4. Sorry, I feel lost as I don’t know your analysis; can it be explained more clearly?

Chapter 4.1.2: Fv/Fm is the maximum quantum yield, a parameter only valid for dark adapted photosynthesis. A decrease indicates chronic photoinhibition or photodamage, a severe harm to the photosynthetic apparatus. Do not confuse with the effective quantum light in light conditions, which shows photoacclimation or light stress effects. It is hard to depress Fv/Fm already by nutrients, as algae do not perform photosynthesis and growth will stop when storage compounds are consumed. Thus, low nutrients level do hardly affect it. Starving plants can have also a high Fv/Fm, but will show a low delta F/Fm’!

Line 509: Of course, a brown community absorbs more light in the blue/green waveband, as already visible in the absorption spectrum, what does not mean that this will be similar to the photosyn-thetic action spectrum. And absorption cross section of a cell is different compared to a community consistent of different cell densities and cell size (e.g influence of sieve effect).

Chapter 4.1.3 Blue light dominates under water, so this may induce rather photoprotection then red light, what your results show. If there is no red light, no “red” photoinhibition can be caused. Photoinhibition is caused by high irradiances, which algae may not encounter in the lower euphotic zone. Only at the water surface this might happen, after longer exposition time. For red light alpha should be higher, as this is similar to shade light conditions. However, the cell does not encounter monochromatic conditions under water, so you need to consider always polychromatic conditions with supporting effects (e.g. Emerson enhancement effect).

Line 635: ETR max you will get only at saturating light conditions, which might not be yet reached by only higher irradiances.

Chapter 4.3.: How do you define photoinhibition? Dynamic photoinhibition is photoprotection and not a damage. Chronic photoinhibition is when reaction center function fails or is damaged. Dynamic photoinhibition/photoprotection you see when the effective quantum yield decrease and chronic when Fv/Fm decreases.
Line 676: light absorption and photochemical quenching, where is this shown in your data?
Line 727: A photoreceptor is a molecule which is not involved in energy transport within the photosynthetic apparatus. Photoreceptors are e.g. phytochroms, cryptochrome etc, which serve for signal transfer/conversion.

Conclusions:
Of course, natural phytoplankton communities can photoacclimate to light quality dynamically under contrasting environmental conditions. Otherwise they would not grow there and would disappear. Your data show that results found in lab experiments with single species are confirmed by field studies and that the blue waveband is the most important abiotic factor for photoacclimation.

Reviewer 3 ·

Basic reporting

The manuscript by Michel-Rodriguez et al. presents an interesting study on the application of multicolor PAM to study the wavelength dependency of natural phytoplankton communities in the English Channel. The manuscipt is quite lengthy but it is generally well-written for a data-intensive manuscript, having the detailed information included to demonstrate its rigour. I think the manuscript is consistent with the scope of PeerJ.

My main critism is that the Abstract is over simplified and does not really reflects the essense of the findings. The authors should provide example of photosynthetic paramaters with values to support the argument/theories especially those related to functional absorption cross section of PSII, which is mentioned in the title but absent in the Abstract. E.g. in "The natural phytoplankton communities ... " (Line 26-28) and in "Two wavelength ratios ..." (Line 30-32).

The authors need to tune down the usage of PETlambda and list out some of actual photosynthetic parameters from time to time throughout the manuscript. I suggest sticking to either PET or PETlambda as there are mixed usage of PET and PETlambda, which can be confusing.

Experimental design

LLine 180: RAMSES has several radiometers. Please provide model and the type of sensors e.g. irradiance, radiance. More information on the deployment of radiometer and data analysis is needed. E.g. How was it deployed? Were both upwelling or downwelling data used and were there binning of the data?

Line 190: You can calculate PAR from the radiometer measurements. Why do you need a separate sensor to measure PAR? Please provide the model and type (e.g. 2pi or 4pi) of the sensor.

Line 220-221: Were all samples subjected to similar dark-adaptation period and at what temperature?

Line 258-260: Can the authors clarify how long it took for each RLC measurement for each sample. With 14 pulses x 20 s x 5 wavelengths, I estimated that each sample will take about 23 minutes to complete. In this case, will the differences in dark-adaptation period affect the output of the photosynthetic parameters?

Line 267: Please provide references to support that PSII and PSI absorb light equally and justify the arbitary factor of 0.5.

Line 305-308: Were the data mean-centred and normalised prior to the PCA analysis?

Ek(II) is an important parameter in the manuscript, however, the information on how it is calculated is missing.

Validity of the findings

Phytoplankton samples were collected at 2 m (Line 157). Were all the sampling stations shallow and well-mixed? How representative are the surface samples for the study area? And how do you relate phytoplankton collected at 2 m with Zeu and Eavg?

Line 193: Surface waves can strongly affect surface PAR (I0) measurements Stramski et al. (2008) and many parameters presented in this manuscript are relying of surface PAR values. Was there any precaution taken or solution to tackle this issue?

Stramski, D., et al. (2008). Relationships between the surface concentration of particulate organic carbon and optical properties in the eastern South Pacific and eastern Atlantic Oceans. Biogeosciences, 5(1):171–201

Line 213: What were the precautions taken to make sure that phytoplankton composition were correctly estimated by the FluoroProbe sensor?

Additional comments

Line 40: "from day to year". The respond of phytoplankton to light can be as fast as minutes if not seconds.

Line 54-56: I think it is a bit inaccurate to cite Schreiber (2004) chapter for PE curves as the chapter is focusing on PAM fluorometry unless the authors were referring to rapid light curves by PAM-fluorometry. I suggest citing conventional PE curves articles e.g. Platt & Jassby (1976).

Line 135: Please provide country of manufacturer.

Line 173: Change "fiberglass" to glass fiber or microfiber

Line 220: What type of net?

Line 245-253: Please provide the equation to derive Sigma(II) and give a brief description of O-I1 especially when mention for the first time.

Line 251: "A" in QA should be subscript.

Section 2.4: As I commented earlier, this is lengthy manuscript. I am not sure if it is necessary to listed all the R packages, especially those used for plotting e.g. biplot (Line 308-309), ggplot2 and ggpur (Line 343-344).

Line 593: "where we collected communities" something is missing here

Figures:
Figure 1: It is not clear to me what is the different between to two lines in each plot.

---

## Round 0.2 · Minor Revisions

The reviewers suggested minor improvements before publication.

Reviewer 1 ·

Basic reporting

No comment

Experimental design

No comment

Validity of the findings

No comment

Additional comments

The authors revised the manuscript appropriately. However, the taxonomic names should be double-checked and corrected as necessary. 'brown-gold' algae is rather a general than a scientific name, therefore e.g. for Nannochloropsis the species name 'Nannochloropsis oculata' and its respective classifications (Eustigmatophyceae) should also be described.

·

Basic reporting

See below general comments

Experimental design

Re-evaluate formula units.

Validity of the findings

no comment

Additional comments

The manuscript is improved and purged from methodical vagueness. However, I still have some question about calculation of Sigma (4) and ETR II (8), see below, which should be cleared.
The authors should also consider, if they would speak from algae in general, this would concern also macroalgae, which as benthic organism would show a different response to light quality and quantity. So it is recommended to speak only from phytoplankton or pelagic microalgae.

Minor comments and corrections are:
Abreviations in Tab.1 are missing and should be included: PE, Pmax, CCA, EC, Zeu,, Zumixl, Eavg, Kd(PAR), O and I1, E avg, DIN, TSS, CDOM. Some are common but even not explained in the text.

Line 84: E is not the abbreviation for irradiance? The response would be named P, or?
Line 202 better: was not uncovered by the waves
Line 212: LI-193 4pi sensor is from LICOR not made by Biosperical Instruments

Line 257: without far red exposure (that would have locked the device for too long with respect to the many measurements required).
Tip: It is not necessary to give a long FR irradiation, to get a significant increase of Fm by release of qE or qT, about 15 s irradiation is generally sufficient. Next time you may check this for measurements. However, in this case with 2 h darkness photosynthesis is already in the dark acclimated state with max. Fm, if no chlororespiration (energetisation of the thylakoids) cause a decrease of Fv/Fm again.

Line 264-269: Here, a part of the old sentence is not deleted, so it is double.
Line 274: the same current fluorescence (F) level of 0.5, here better to write Ft
Line 285 ff: this is misunderstanding τ must be the time needed to reach the I1-level, otherwise the formula is meaningless, as you have always irradiated for 1ms. According to your text τ would be always 1ms.
Line 287: Avogadro's constant (6.022.1023 quanta.(mol photons)-1), are the units right? quanta or photons? I suppose you have to delete quanta. Shouldn´t be Sigma dimensionless or m-1? Please control units.
Line 290f: there is some repetition to the sentences above.
Formula 8: It is curiously as you divide by L for determination of Sigma and multiply it again for PAR II
Why?

Line 564: increase in the “green” absorption capacity of fucoxanthin, Not better due to fucoxanthin,? as the molecule itself will not change its absorption characteristic.

Line 626ff: you should write brown microalgae or phytoplankton because in multicellular macroalgae you will observe different result. Generalization is here not appropriate. The same for the algae groups mentioned later, because benthic macro- or microalgae may respond differently.

Line 850. different , typing error.

Supplements:
Fig 4s : Labeling of axis is missing, figure not complete

Reviewer 3 ·

Basic reporting

No comment

Experimental design

No comment

Validity of the findings

No comment

Additional comments

No comment

---

## Round 0.3 · accepted · Accept

Your manuscript is accepted for publication

·

Basic reporting

All misunderstandings an errors are resolved.
The publication is from my view ready for publication.

Experimental design

All misunderstandings resolved

Validity of the findings

See above